# Insula to mPFC reciprocal connectivity differentially underlies novel taste neophobic response and learning in mice

Haneen Kayyal[1], Sailendrakumar Kolatt Chandran[1], Adonis Yiannakas[1], Nathaniel Gould[1], Mohammad Khamaisy[1], Kobi Rosenblum[1,2]*

[1]Sagol Department of Neuroscience, University of Haifa, Mount Carmel, Israel; [2]Center for Gene Manipulation in the Brain, University of Haifa, Mount Carmel, Israel

**Abstract** To survive in an ever-changing environment, animals must detect and learn salient information. The anterior insular cortex (aIC) and medial prefrontal cortex (mPFC) are heavily implicated in salience and novelty processing, and specifically, the processing of taste sensory information. Here, we examined the role of aIC-mPFC reciprocal connectivity in novel taste neophobia and memory formation, in mice. Using pERK and neuronal intrinsic properties as markers for neuronal activation, and retrograde AAV (rAAV) constructs for connectivity, we demonstrate a correlation between aIC-mPFC activity and novel taste experience. Furthermore, by expressing inhibitory chemogenetic receptors in these projections, we show that aIC-to-mPFC activity is necessary for both taste neophobia and its attenuation. However, activity within mPFC-to-aIC projections is essential only for the neophobic reaction but not for the learning process. These results provide an insight into the cortical circuitry needed to detect, react to- and learn salient stimuli, a process critically involved in psychiatric disorders.

*For correspondence:
kobir@psy.haifa.ac.il

Competing interests: The authors declare that no competing interests exist.

## Introduction

Determining salient stimuli from the continuously perceived sensory input is a key component for effective learning that has been shaped throughout evolution. The term 'salient' is used in psychology and neuroscience to denote a particularly distinct or important stimulus, or an aspect of it (*Yantis and Hillstrom, 1994*). Attributing salience to a given cue or stimulus, can be affected by genetics, previous experiences, current psychological and motivational state, stress response, goals and motivation (*Puglisi-Allegra and Ventura, 2012*; *Goldberg et al., 2006*). Numerous studies have identified cortical and subcortical brain structures that are involved in the detection of salient stimuli, with the insular cortex (IC) being a key node of the salience network (*Gogolla, 2017*; *Livneh et al., 2017*; *Uddin, 2015*). In particular, the IC is believed to have a critical role in the bottom-up detection of salient events and attention allocation (*Menon and Uddin, 2010*). Thus, when a salient stimulus is detected, the insula assists in targeting other brain structures that enable access to attention and working memory resources.

Salient experiences are comprised of several dimensions, which include novelty amongst others (*Modinos et al., 2015*; *Schultz, 2013*; *Winton-Brown et al., 2014*). Experiencing a new taste therefore represents a salient experience of novel sensory information, and the memory for this event is dependent on the functionality of the IC (*Gal-Ben-Ari and Rosenblum, 2011*; *Yiannakas and Rosenblum, 2017*). Taste recognition, upon which novelty and salience detection inversely depend, relies on a combination of instinctive responses (genetic elements) and brain processes involved in recalling past experiences to help animals cautiously avoid poisonous food, while taking advantage of food that is deemed safe. Therefore, when animals encounter a taste that is completely new to

them, they approach it with cautiousness, a phenomenon called taste neophobia (*Lin et al., 2015*). This neophobic response is the first line of defense, and therefore an important and evolutionarily conserved behavior, manifesting in reluctance to consume the novel taste, for fear of potentially adverse outcomes. Following this tentative consumption of the novel taste, and in the absence of any proceeding aversive visceral consequences, a memory for a safe taste is formed (*Osorio-Gómez et al., 2018*). Thus, the consumption of this taste gradually increases in the following encounters, a phenomenon called attenuation of neophobia (*Barot and Bernstein, 2005*; *Green and Parker, 1975*; *Morón and Gallo, 2007*). This is an example in which a salient, novel stimulus (1) increases attention which affects the preliminary response, and then (2) initiates incidental encoding for a memory of the new experience as to subsequently prime future adaptive behavior (*Borsook et al., 2007*; *Anderson et al., 2006*; *McGaugh, 2006*). Thus, acting when the sensory information is new, and then learning the new stimulus (familiarization), are both crucial but different components in the process of behavior to a salient, novel stimulus.

The process by which a novel taste becomes a familiar one is partially subserved by the IC (*Bermudez-Rattoni, 2014*), and the gustatory cortex, located within the medial portion of the anterior IC (aIC) (*Gehrlach et al., 2020*; *Shura et al., 2014*; *Rolls, 2016*; *Yiannakas and Rosenblum, 2017*). The aIC plays a critical role in the formation and retrieval of novel taste memory (*Kayyal et al., 2019*; *Lavi et al., 2018*; *Yiannakas and Rosenblum, 2017*). Inhibition of protein synthesis or lesions to the IC cause impairment in novel taste learning (*Rosenblum et al., 1993*). Several molecular changes occur in the IC following novel taste learning (*Gal-Ben-Ari and Rosenblum, 2011*; *Gould et al., 2020*), including increased phosphorylation-activation of the extracellular regulated kinase (ERK). Phosphorylated ERK (pERK) levels are increased in the IC following novel taste consumption (*Berman et al., 1998*) and inhibition of MEK, an upstream kinase of ERK in the IC, impairs novel taste learning (*Berman et al., 1998*).

Although an accumulating body of work has been directed toward understanding the aIC and the molecular mechanisms by which novel taste memory is formed and retrieved (*Adaikkan and Rosenblum, 2015*; *Merhav et al., 2006*; *Yefet et al., 2006*), the cellular and circuit mechanisms are yet to be deciphered.

Additionally to the aIC, the mPFC is another region that plays a prominent component of the novelty network that is engaged following an encounter with a novel taste stimulus (*Morici et al., 2015*; *Takehara-Nishiuchi et al., 2020*). Lesions of the mPFC result in impaired learning of novel tasks (*Barker et al., 2007*; *Devito and Eichenbaum, 2011*), including taste learning (*Gonzalez et al., 2015*). The mPFC is reciprocally connected with the IC, however, whether and how their connectivity contributes to novel taste learning has not been comprehensively examined (*Gabbott et al., 2003*). Hence, we set out to assess the involvement of aIC-mPFC functional connectivity in the neophobic response evoked during novel taste exposure, as well as novel taste learning. To do so, we measured the correlation between the number of activated, pERK$^+$ neurons within the general aIC and mPFC populations, and specifically within aIC-to-mPFC or mPFC-to-aIC projecting neurons. We found that experiencing a novel taste causes a significant increase in reciprocal aIC-mPFC neuronal activity as measured by pERK$^+$ neurons (*Jones et al., 1999*). From an electrophysiological perspective, we identify a correlation between taste novelty and increased excitability in aIC-to-mPFC projecting neurons. In order to ascertain behavioral causality of the molecular and electrophysiological correlations we identified, we inhibited the activity of each pathway during the reaction to- or the learning of- a novel taste. We found that chemogenic inhibition of the aIC-to-mPFC pathway resulted both in impaired neophobic response, as well as impaired taste learning and memory. In contrast, inhibition of the opposite, reciprocal, mPFC-to-aIC pathway, caused impairment to the neophobic response only, but not to the learning process.

Our results show that activation of the aIC-mPFC circuit is correlative and necessary for both the neophobic response to novel tastants, as well as memory formation, with functionally discrete reciprocity. This provides part of a functional map of the broader salience system that is yet to be described, as well as providing a guideline for setting the existing molecular knowledge to reveal the unexplored circuit framework.

# Materials and methods

## Animals

Animals used were 8-to 12-week-old wild type (WT) adult (C57BL/6) male mice. Mice were kept in the local animal resource unit at the University of Haifa in an environment that is temperature-controlled and under a 12 hr dark/light cycle. Water and chow pellet were available ad libitum. All experiments and procedures conducted were approved by the University of Haifa Animal Care and Use committee under Ethical license 554/18 and were in accordance with the National Institutes of Health guidelines for ethical treatment of animals.

## Surgery and viral injections

For all surgeries, naive adult male mice were used as detailed. Mice received i.p. injections of norocarp (0.5 mg/kg), 30 min before surgery and 24 hr later, to minimize pain. Animals were anesthetized by i.p. injection of ketamine and Dormitor (0.5 mg/kg) and then placed in a stereotaxic device (a Model 963 Kopf Instruments stereotaxic injection system). After exposing the scalp, and using bregma and lambda as relevant alignment points, holes were drilled (0.4 nm) in both hemispheres. Mice were injected with AAV construct at the IC (A-P: +0.86; D-V-4; M-L: ±3.4) and/or the mPFC (A-P: 2.34; D-V: −2; M-L:±0.3) or the basolateral amygdala (BLA) (A-P: −1.6; D-V-4.8; M-L: ±3.375), as stated in each experiment. All AAV constructs used in this study were obtained from the viral vector facility of the University of Zurich (http://www.vvf.uzh.ch). All mice used in our studies were injected with 0.25 ul/site/hemisphere of said AAV constructs (physical titer: 4.5 x 10E12 vg/ml) as defined in each experiment. Viral constructs were injected at a rate on 0.1 µl/min, while the syringe was kept in the injection site for 5 min prior and 10 min following delivery, to ensure efficient distribution of the virus. The skin was closed using Vetbond glue, and mice were given 4 weeks of recovery to guarantee viral expression. After recovery, mice were split into individual cages for behavioral experimental purposes.

## Immunohistochemistry and quantification

For perfusion, mice that underwent behavioral experimentation were deeply anesthetized using isoflurane. When the mice were completely anesthetized, they were perfused transcardially using 4% paraformaldehyde (PFA) in 0.1M PBS solution (PBS, MFCD00131855, Sigma-Aldrich).

After perfusion, brains were placed in falcon tubes containing 4% formaldehyde solution overnight at 4°C. On the following day, the solution was replaced with 30% sucrose in 0.1 M phosphate buffered saline (PBS) and brains were incubated for another 48 hr. Afterwards, brains were stored at −80°C. Coronal sections (40 mm thickness) were then collected from the IC (Bregma: 1.10 mm to 0.26 mm) or the mPFC (Bregma 1.94 mm to 1.54 mm) utilizing a Leica cryostat CM 1950. Following sectioning, slices were washed using PBS for three times, and then blocked for 1 hr using a 0.3% bovine serum albumin (10775835001, Sigma-Aldrich), 0.3% triton X-100 (MFCD00128254, Sigma-Aldrich) and 10% fetal bovine (MFCD00132239, Sigma-Aldrich) solution in PBS. After washing three times with PBS, slices were incubated at 4°C with primary antibodies against rabbit phospho-ERK (1:200 D13.14.4E Cell Signaling) and mouse NeuN (1:500 MAB377 MERCK) in blocking solution. On the next day, the primary antibody solution was removed, and slices were washed three times with PBS. Then, slices were kept in the dark in blocking solution containing donkey anti-rabbit Alexa Fluor 647 (1:500 ab150075 Invitrogen) and Goat anti-mouse Alexa Fluor 488 (1:500 A11001 ThermoFisher) secondary antibodies, for 1.5 hr. Slices were then washed three times using PBS, and mounted on glass slides and Vectashield mounting solution containing DAPI (H-1200) was added prior to adding coverslips.

Slices were visualized and images were acquired using a vertical light microscope (Olympus Cell-Sens Dimension ) using ×10 or ×20 magnification. pERK data was subsequently quantified and analyzed in terms of the bilateral number of positive cells/slice using two-way ANOVA (Graphpad Prism ), to examine the respective responses in the general population. The number of rAAV+ cells was similarly quantified and analyzed in the above slices. The images of IC and/or mPFC slices were processed with Image-Pro Plus V-7, Media Cybernetics , pERK+ and rAAV+ neurons were quantified manually in the relevant subregions and layers. To assess the recruitment of the IC-to-mPFC or mPFC-to-IC projecting neurons in the pERK population, the number of cells co-localized with rAAV+ cells was

normalized to the total rAAV$^+$ population [(rAAV$^+$pERK$^+$/ rAAV$^+$)*100%] in the respective subregions and layers in slices of the IC or the mPFC. Quantification was done using randomly assigned IDs for individual animals, regardless of treatment. Following quantification, we confirmed treatments, and data were analyzed (Graphpad Prism ).

## Behavioral experiments

Mice were randomly allocated to experimental groups. Group size range estimation was based on previously published results using similar methods, as well power calculations (https://www.stat.ubc.ca/~rollin/stats/ssize/n2.html).

### Immunohistochemical studies

Mice aged 8–12 weeks were injected with rAAV-hSyn1-chl-mCherry-WPRE-SV40p(A), rAAV construct, at the mPFC (to label IC-to-mPFC projecting neurons) or at the IC (to label mPFC-to-IC projecting neurons). One month later, animals were individually separated and given 5 acclimation days. Next, mice were water deprived for 24 hr, and were given tap water (for novel saccharin group) or saccharin (0.5% dissolved in tap water, for familiar saccharin group) pipettes for 20 min sessions each day for 6 consecutive days. On the 7th day, all animals were presented with 1 ml of saccharin, and perfused 20 min later, for immunohistochemical analysis.

### IC-to-mPFC/ mPFC-to-IC circuit inhibition during novel saccharin exposure, or familiar taste retrieval

In order to target the IC-to-mPFC circuit for inhibition, mice aged 8–12 weeks were bilaterally injected with AAV8_hEF1a-dlox-hM4D(Gi)_mCherry(rev)-dlox-WPRE-hGHp(A), a Cre-dependent AAV construct expressing the inhibitory DREADD receptor in neurons of the IC. Bilateral injection in the mPFC with rAAV-hSyn1-chl-EGFP_2A_iCre-WPRE-SV40p(A), a retrograde-Cre AAV construct, was also conducted, as to target IC-to-mPFC projecting neurons for inhibition. As a control study, another set of mice were injected with AAV8_hEF1a-dlox-mCherry (rev)-dlox-WPRE-hGHp (A) at the aIC, an mCherry construct without DREADD receptor, while the aforementioned retrograde-Cre AAV construct was injected at the mPFC.

To target mPFC-to-IC projecting neurons for inhibition, the same constructs were used to injections in a different set of mice, with the retrograde-Cre AAV construct being injected in the IC, and the Cre-dependent DREADD injected at the mPFC.

Animals were then given 4 weeks of recovery. Subsequently, animals were individually separated and given 5 days for acclimation. Mice were then water deprived for 24 hr, and given 20 min of pipettes containing tap water for 20 min session for 3 days. Next, animals were i.p. injected with CNO (diluted in saline; 0.5 mg/kg; Enzo) or Saline for control (1% body weight) 1 hr prior to a choice test. Choice test of two pipettes was provided to the mice, with one pipette containing 0.5% saccharin (their first exposure to saccharin) and the other containing water. Drinking volumes for each solution were measured and aversion index scores were calculated (aversion index = volume of water consumed/volume of (water + saccharin) consumed%).

During the subsequent 4 days, mice were presented with unreinforced choice tests of saccharin and water, for 20 min session each day, without any intervention. On the 6th day, animals were only given water in pipettes. Toward examining the importance of the circuitry during familiar taste retrieval, mice were injected on the 7th day with CNO or saline, 1 hr prior to an additional choice test.

### Circuit inhibition during aversive taste memory retrieval

In order to test the importance of IC-to-mPFC projecting neurons in aversive taste memory retrieval, animals were injected with AAV8_hEF1a-dlox-hM4D(Gi)_mCherry(rev)-dlox-WPRE-hGHp(A), a Cre-dependent AAV construct expressing the inhibitory DREADD receptor at the IC, and rAAV-hSyn1-chl-EGFP_2A_iCre-WPRE-SV40p(A), a retrograde AAV Cre construct at the mPFC. 4 weeks after the surgery, mice were individually housed and allowed 5 days for acclimation. Mice were then water deprived for 24 hr prior to being allowed access to water pipettes in 20 min sessions for 3 days. On the following day, mice were trained in CTA by exposing them to saccharin for the first time. Forty min after saccharin consumption, mice were i.p. injected with a 2% body weight dose of LiCl (0.14

M), a malaise inducing agent. Mice were water-restricted for the subsequent 2 days. On the 4th day, mice were injected with CNO or saline, 1 hr before saccharin/water choice test. After 20 min of choice test, drinking volumes were measured and aversion index scores were calculated as above.

### Circuit inhibition during innately aversive novel taste exposure

In order to target IC-to-mPFC projecting neurons for inhibition, mice aged 8–12 weeks were bilaterally injected with AAV8_hEF1a-dlox-hM4D(Gi)_mCherry(rev)-dlox-WPRE-hGHp(A), a Cre-dependent AAV construct expressing the inhibitory DREADD receptor in neurons of the IC, and bilateral injection of rAAV-hSyn1-chI-EGFP_2A_iCre-WPRE-SV40p(A), a retrograde-Cre AAV construct, at the mPFC. Animals were individually separated a month after the injection and given 5 days of acclimation. Mice were then water deprived for 24 hr and were given water pipettes for 20 min each day for 3 days. On the test day, mice were injected with CNO or saline one hour prior to a choice test between water and Quinine (0.04%). The mice were presented with the choice test for 20 min and drinking volumes of each solution was measured. Aversion was calculated accordingly.

### IC-to-BLA circuit inhibition during expression of neophobia

In order to target the IC-to-BLA circuit for inhibition, mice aged 8–12 weeks were bilaterally injected with AAV8_hEF1a-dlox-hM4D(Gi)_mCherry(rev)-dlox-WPRE-hGHp(A), a Cre-dependent AAV construct expressing the inhibitory DREADD receptor in neurons of the IC. Bilateral injection in the BLA with rAAV-hSyn1-chI-EGFP_2A_iCre-WPRE-SV40p(A), a retrograde-Cre AAV construct, was also conducted, as to target IC-to-BLA projecting neurons for inhibition. Animals were then given 4 weeks of recovery, prior to being individually housed and given 5 days for acclimation. Then, animals were water deprived for 24 hr, and given 20 min of pipettes containing tap water for 20 min session for 3 consecutive days. Next, animals were i.p. injected with CNO (0.5 mg/kg; Enzo) or Saline for control (1% body weight) 1 hr prior to a choice test. Choice test of two pipettes was provided to the mice, with one pipette containing 0.5% saccharin (their first exposure to saccharin) and the other containing water. Drinking volumes were measured and aversion was calculated accordingly.

During the subsequent 4 days, mice were presented with unreinforced choice tests of saccharin and water, in 20 min sessions each day, without any additional intervention.

### IC-to-mPFC/ mPFC-to-IC circuit inhibition during an open field test

Mice aged 8–12 weeks were bilaterally injected with AAV construct expressing the inhibitory DREADD receptor in IC-to-mPFC or mPFC-to-IC projecting neurons as previously described. As a control study, another set of mice were injected with a control Cre-dependent mCherry-expressing AAV vector at the aIC and a retrograde-Cre AAV construct at the mPFC. All three groups were i.p. injected with CNO, and 1 hr later, they were placed in a 50×50 cm open-field arena (Noldus Information Technology), for 10 min.

## Electrophysiology

To label neurons in the IC that project to the mPFC, mice were injected with rAAV-hSyn1-chI-mCherry-WPRE-SV40p(A) construct, at the mPFC. One month after the surgery, mice were split to individual cages, and given 5 days of acclimation period. Mice were water deprived for 24 hr, and then presented with water (for novel saccharin group) or saccharin (for familiar saccharin group) for six consecutive days. On the 7th day, mice were presented with 1 ml of saccharin and sacrificed 1 hr later. A cage control group, injected with the rAAV-hSyn1-chI-mCherry-WPRE-SV40p(A) construct at the mPFC, were also sacrificed without any separation nor water restriction procedure. To obtain brain slices containing the aIC, mice were deeply anesthetized with 5%isoflurane and transcardially perfused with 40 ml of ice-cold oxygenated cutting solution containing the following (in mM): 25 NaHCO3, 105 Choline-Chloride, 2.5 KCl, 7 MgCl2, 0.5 CaCl2, 1.25 NaH2PO4, 25 D-glucose, 1 Na-Ascorbate and 3 Na-Pyruvate. All reagents were commercially obtained from Sigma-Aldrich Israel, except where stated. The 300-μm-thick coronal brain slices were cut with a Campden-1000 Vibrotome using the same cutting solution. The slices were allowed to recover for 30 min at 37°C in artificial CSF (ACSF) containing the following (in mM): 125 NaCl, 2.5 KCl, 1.25 NaH2PO4, 25 NaHCO$_3$, 25 D-glucose, 2 CaCl$_2$, and 1 MgCl$_2$, followed by additional recovery for at least 30 min in ACSF at

room temperature until electrophysiological recording. The solutions were constantly gassed with carbogen (95% $O_2$ 5% $CO_2$).

## Intracellular whole-cell recording

After the recovery period, slices were placed in the recording chamber and maintained at 32–34°C with continuous perfusion of carbogenated ACSF (2 ml/min). Brain slices containing the aIC were illuminated with infrared light and pyramidal cells were visualized under a differential interference contrast microscope with 10X or 40X water-immersion objectives mounted on a fixed-stage microscope (BX51-WI; Olympus). The image was displayed on a video monitor using a charge-coupled device (CCD) camera (Dage MTI). aIC-to-mPFC projecting neurons were identified by mCherry, and whole cell recordings from aIC to mPFC-projecting neurons were performed using an Axopatch 200B amplifier and digitized by Digidata 1440 (Molecular Devices). The recording electrode was pulled from a borosilicate glass pipette (3–5 MΩ) using an electrode puller (P-1000; Sutter Instruments) and filled with a K-gluconate-based internal solution (in mM): 130 K-gluconate, 5 KCl, 10 HEPES, 2.5 MgCl2, 0.6 EGTA, 4 Mg-ATP, 0.4 Na3GTP and 10 phosphocreatine (Na salt). The osmolarity was 290 mOsm, and pH was 7.3. The recording glass pipettes were patched onto the soma region of rAAV$^+$ pyramidal neurons.

The recordings were made from the soma of aIC pyramidal cells, particularly from layer 2/3 and Layer 5/6. Liquid junction potential (10 mV) was not corrected online. All current clamp recordings were low-pass filtered at 10 kHz and sampled at 50 kHz. Series resistance was compensated and only series resistance <20 MΩ was included in the dataset. Pipette capacitance was ~ 80% compensated. The method for measuring active intrinsic properties was based on a modified version of previous protocols (*Kaphzan et al., 2013*; *Sharma et al., 2018*).

## Recording parameters

Resting membrane potential (RMP) was measured 10 s immediately after the beginning of whole cell recording (rupture of the membrane under the recording pipette). The dependence of firing rate on the injected current was obtained by injection of current steps (of 500 ms duration from +50 to 400 pA in 50 pA increments). Input resistance (Rin) was calculated from the voltage response to a hyperpolarizing current pulse (−150 pA). Sag ratio was calculated from voltage response −150 pA. The sag ratio during the hyperpolarizing steps calculated as $[(1-\Delta V_{SS}/ \Delta V_{max}) \times 100\%]$ as previously reported (*Song et al., 2015*). Membrane time constant was determined using a single exponential fit first 100 ms of raising phase of cell response to 1 s −150 pA hyperpolarization step.

For measurements of a single action potential (AP), after initial assessment of the current required to induce an AP at 15 ms from the start of the current injection with large steps (50 pA), a series of brief depolarizing currents were injected for 10 ms in steps of 10 pA increments. The first AP that appeared on the 5 ms time point was analyzed. A curve of dV/dt was created for that trace and the 30 V/s point in the rising slope of the AP was considered as threshold (*Sharma et al., 2018*). AP amplitude was measured from the equipotential point of the threshold to the spike peak, whereas AP duration was measured at the point of half-amplitude of the spike. The medium after-hyperpolarization (mAHP) was measured using prolonged (3 s), high-amplitude (3 nA) somatic current injections to initiate time-locked AP trains of 50 Hz frequency and duration (10–50 Hz, 1 or 3 s) in pyramidal cells. These AP trains generated prolonged (20 s) AHP, the amplitudes and integrals of which increased with the number of APs in the spike train. AHP was measured from the equipotential point of the threshold to the anti-peak of the same spike (*Gulledge et al., 2013*). Series resistance, Rin, and membrane capacitance were monitored during the entire experiment. Changes 30% in these parameters were criteria for exclusion of data.

# Results

## Novel taste experience increases the number of pERK$^+$ cells in mPFC-projecting neurons from inner layers of the aIC

Activity in the IC is correlated and necessary for the formation of taste memory traces (*Gal-Ben-Ari and Rosenblum, 2011*). The IC displays high connectivity with other brain regions, and these reciprocal interactions facilitate integration of sensory information, valence and reward

(*Yiannakas and Rosenblum, 2017*). Previously, we identified a correlation between an aversive taste valence and activation of aIC-to-BLA projecting neurons (*Lavi et al., 2018*). We also proved that activity of these projections is necessary for the acquisition and retrieval of an aversive taste memory, but not for attenuating a neophobic response to a novel, non-aversive taste (*Kayyal et al., 2019*). Given that mPFC activity is crucial for novel taste memory formation (*Gonzalez et al., 2015*; *Uematsu et al., 2015*; *Mickley et al., 2007*) and is correlated with learning novel experiences (*Euston et al., 2012*), we examined the role of aIC-mPFC reciprocal projections in novel taste learning. Toward that end, WT mice were injected with a retrograde virus containing an mCherry construct in the mPFC, labeling neurons in the IC that project to the mPFC (*Figure 1i*). As ERK activation is necessary for taste learning (*Elkobi et al., 2008*), we used pERK, as an indicator of activated neurons (*Sweatt, 2001*; *Adaikkan and Rosenblum, 2012*). We quantified the number of pERK$^+$ neurons in 24 aIC slices (bregma 1.10 mm- 0.26 mm) of these mice following novel (one exposure, n=3) or familiar (eight exposures, n=5) taste experiences involving the same taste, saccharin (see Materials and methods, *Figure 1a,h*). Quantification of pERK$^+$ cells across the aIC demonstrated novel saccharin consumption to be associated with an increased number of activated, pERK$^+$ neurons compared to familiar saccharin, in agreement with the literature (*Berman et al., 1998*) (unpaired t-test: p=0.0002, t=3.883, DF=114; *Figure 1b*). The aIC itself is comprised of three major subregions: the agranular (AIC), dysgranular (DIC), and granular IC (GIC) with the 4$^{th}$ cortical layer persisting only in the dorsal portion of the GIC (Figure 7). Within these subregions, experiencing novel saccharin increased the number of pERK$^+$ neurons in the AIC (p=0.04, t=2.094, DF=67; *Figure 1c*) and the DIC (unpaired t-test: t=3.749, p=0.0003, DF=94; *Figure 1d*), but not the GIC (unpaired t-test: t=1.340, p=0.184, DF=92; *Figure 1e*) compared to familiar saccharin. Different cortical layers play different roles in sensory information processing (*Dikecligil et al., 2020*; *Harris and Mrsic-Flogel, 2013*). Hence, we examined the expression of pERK in the outer (1,2,3 combined) and inner (5,6 combined) layers. We found that both in outer (unpaired t-test: p=0.0002, t=3.897, DF=70; *Figure 1f*) and inner layers (unpaired t-test; p=0.002, t=3.121, DF=89; *Figure 1g*) of the entire aIC, there is a significant increase in pERK$^+$ neurons following novel saccharin experience, compared to the familiar saccharin group. Extensive reciprocal connectivity exists between the IC (mainly the AIC and DIC) and the mPFC (*Gabbott et al., 2003*), and this may be analogous to the salience network in humans (*Uddin, 2015*). We therefore hypothesized that aIC-to-mPFC projections have a vital role in novel taste behavior. We quantified pERK$^+$ neurons in the AIC and DIC that co-localized with retrograde-virus labeling, originating from the mPFC. We found a significant increases in the percentage of mPFC-projecting aIC neurons that are pERK$^+$ following consumption of a novel-compared to familiar taste (unpaired t-test: t=3.301, p=0.0015, DF=69; *Figure 1j*). This effect was evident at the AIC (unpaired t-test: t=3.513, p=0.0008, DF=69; *Figure 1k*) but not the DIC (unpaired t-test: t=1.353, p=0.181, DF=70; *Figure 1l*). Looking at the cortical layer distribution, we found that pERK induction in the aIC is observed in both the inner layers (unpaired t-test: t=3.050, p=0.003, DF=92; *Figure 1n*) and outer layers of the aIC (Mann Whitney test: p=0.0069, U=352; *Figure 1m*). To test for variability of labeling between novel/familiar saccharin groups, we perfromed an analysis of number of the aIC rAAV$^+$ neurons in its different subregions/layers and found no significant difference in number of rAAV$^+$ neurons between the groups (*Figure 1—figure supplement 1a,b*).

## Novel taste experience increases excitability in mPFC-projecting neurons of the inner layers of the aIC

Following the correlation between experiencing a novel taste and increased activation of aIC-mPFC projecting neurons as measured by pERK (*Figure 1*), considering previous evidence regarding the role of pERK on neuronal intrinsic properties (*Cohen-Matsliah et al., 2007*), we tested the hypothesis that following the experience of a novel taste, aIC-to-mPFC projecting neurons will become more excitable. Toward that end, we once more injected a rAAV mCherry construct in the mPFC (*Figure 2b*), allowing the labeling of mPFC-projecting neurons of the IC. A month later, different groups of mice were exposed to novel (n=four mice) or familiar saccharin (n=four mice) and were sacrificed one hour later (see materials and methods, *Figure 2a*) while cage control mice were sacrificed without water restriction regime (n=4). Whole-cell patch-clamp recordings were then performed in slices from aIC-to-mPFC-projecting neurons (*Figure 2b*). In accordance with our hypothesis, novel taste consumption increased the excitability of mPFC-projections in the inner

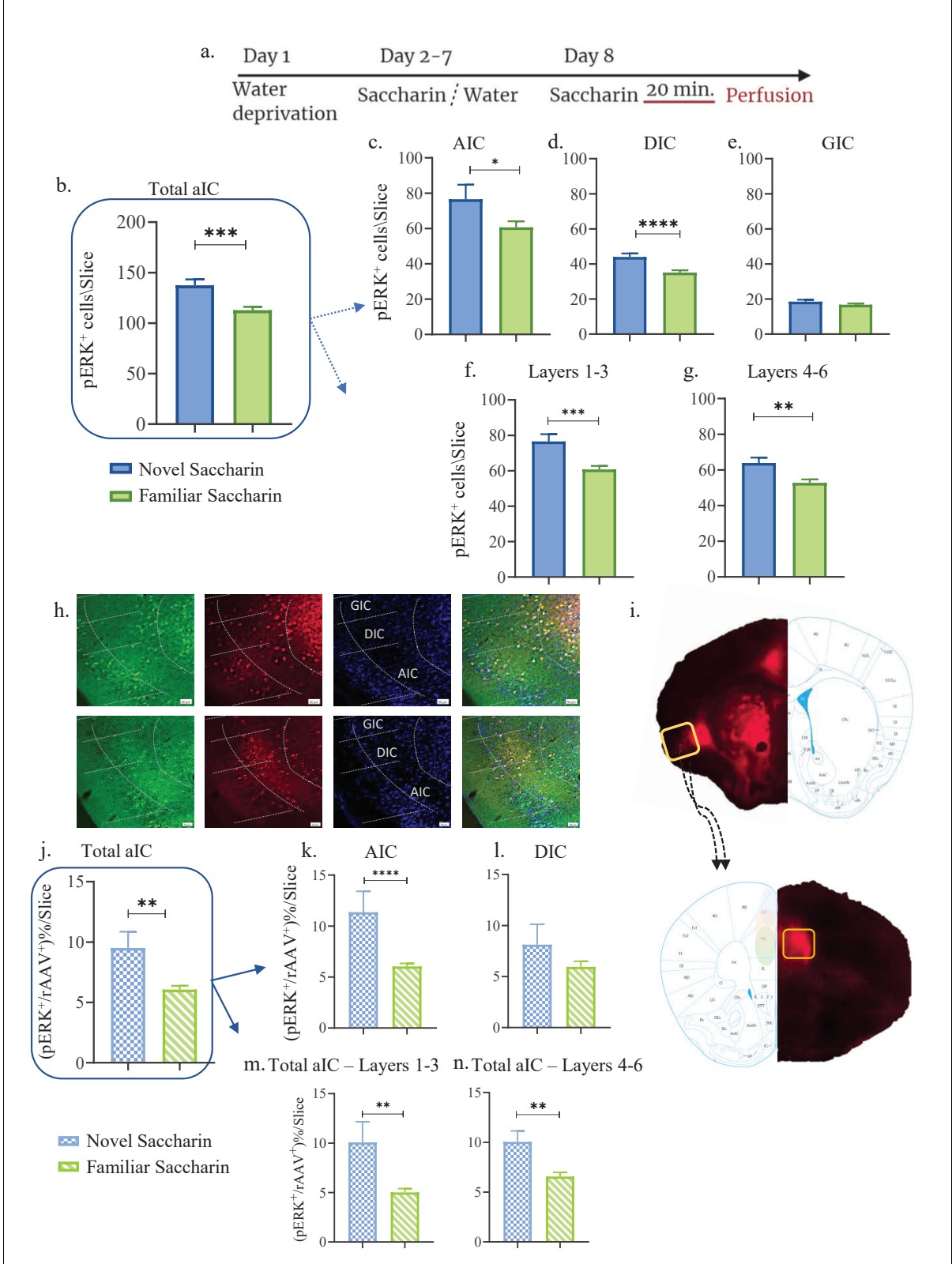

**Figure 1.** Novel taste experience increases number of pERK+ cells in mPFC-projecting aIC neurons. (a) Schematic representation of novel or familiar taste learning; animals were water restricted in the first day and administered with saccharin (familiar saccharin group) or water (novel saccharin group) in the following 6 days. In the last day, mice were presented with 1 ml of Saccharin 20 min prior to perfusion. Number of pERK+ neurons was quantified in the different subregions and layers of the IC. (b) Number of pERK+ neurons was significantly increased in the total IC following novel (137.4±5.876)

*Figure 1 continued on next page*

*Figure 1 continued*

compared to familiar (112.9±3.308) saccharin consumption. (**c**) Number of pERK⁺ neurons was significantly increased in the AIC following novel (76.6±8.201) compared to familiar (60.62±3.476) saccharin consumption. (**d**) Number of pERK⁺ neurons was significantly increased in the DIC following novel (44±1.996) compared to a familiar (35.13±1.270) saccharin consumption. (**e**) Number of pERK⁺ neurons was unchanged in the GIC following novel (18.5±1.096) and familiar (16.79±0.6797) saccharin consumption. (**f**) Number of pERK⁺ neurons was significantly increased in the outer layers of the IC following novel (76.58±4.103) compared to familiar (60.75±2.016) saccharin consumption. (**g**) Number of pERK⁺ neurons was significantly increased in the inner layers of the IC following novel (63.87±3.009) compared to familiar (52.76±1.868) saccharin consumption. (**h**) Representative coronal IC sections immunostained for pERK (green) and DAPI (blue) from mice injected with retroAAV at the mPFC (red) following novel (upper) and familiar (lower) saccharin. Scale bar, 50 µm, 20x. (**i**) Stereotaxic injection of rAAV-mCherry construct (red) at the mPFC and its subsequent labeling in the IC. Representative schematic overlays of the Cre-dependent expression of the chemogenetic receptors using the rAAV systems is shown, demonstrating the expressionto be restricted in the aIC and mPFC. Number of double-labeled (pERK⁺, rAAV⁺) neurons was calculated as a percentage of all rAAV⁺ neurons. (**j**) Percentage of double-labeled neurons of the IC was significantly higher following novel (9.517±1.337) compared to familiar (6.065 ± 0.313%) saccharin consumption. (**k**) Percentage of double-labeled neurons of the AIC was significantly higher following novel (11.36 ± 2.050%) compared to familiar (6.036±0.303) saccharin consumption. (**l**) Percentage of double-labeled neurons of the DIC was similar following novel (8.126 ± 2%) and familiar (5.956 ± 0.546%) saccharin consumption. (**m**) Percentage of double-labeled neurons of the outer layers of the IC was similar following novel (10.08 ± 2.079%) and familiar (5.02 ± 0.381%) saccharin consumption. (**n**) Percentage of double-labeled neurons of the inner layers of the IC was similar following novel (10.05 ± 1.089%) and familiar (6.562 ± 0.411%) saccharin consumption. Data are shown as mean ± SEM. *p<0.05, **p< 0.01, ***p<0.001, ****p<0.0001.

The online version of this article includes the following source data and figure supplement(s) for figure 1:

**Source data 1.** Novel taste experience increases number of pERK cells in mPFC-projecting aIC neurons.

**Figure supplement 1.** aIC-mPFC reciprocal connectivity is prominent in the AIC and DIC of the aIC and the prelimbic subregion of the mPFC.

**Figure supplement 1—source data 1.** aIC-mPFC reciprocal connectivity is prominent in the AIC and DIC of the aIC and the prelimbic subregion of the mPFC.

layers of the aIC (*Figure 2c–d*), but not the outer layers (*Figure 2—figure supplement 1a–j*). Intrinsic property changes included RMP (*Figure 2—figure supplement 2c*) and AP half -width (*Figure 2—figure supplement 2e*), with depolarized Rin being observed following novel and familiar saccharin consumption, while AP half –width was significantly reduced following novel saccharin consumption compared to the cage controls. In addition, Rin was increased following novel experience compared to the familiar saccharin consumption (*Figure 2—figure supplement 2h*). Other intrinsic properties, including rheobase, mAHP, firing threshold, time constant and AP amplitude did not show any differences between the groups (*Figure 2—figure supplement 2a–j*). Interestingly, mPFC-projecting neurons of the outer layers of the aIC did not exhibit any significant differences in their intrinsic properties when similarly comparing novel (n=three mice) and familiar (n=four mice) tastants (*Figure 2—figure supplement 2a–j* and *Table 1*).

Taken together, these results demonstrate that following the salient experience of novel taste consumption, deep layer IC-to-mPFC neurons are activated and display increased excitability, thus identifying part of a circuit involved in this form of learning.

## Novel taste experience increases activation of aIC-projecting neurons of the mPFC

The mPFC is involved in novelty behavior and the connectivity between the IC and the mPFC is reciprocal (*Gabbott et al., 2003*; *Gehrlach et al., 2020*). Considering the present results, we sought to investigate the role of mPFC inputs to the aIC in novel taste learning. Hence, a separate group of mice were injected with the rAAV mCherry-expressing construct in the aIC, labeling neurons in the mPFC that project to the IC (*Figure 3h,i*). Mice were exposed to novel (first exposure, n=3, *Figure 3a*) and familiar (eight exposures, n=3) saccharin. As previously done in the IC, the general population of pERK⁺ neurons were evaluated in the different mPFC subregions - prelimbic (PrL), infralimbic (InfrL), and cingulate cortex (Cg) from eight slices (Bregma 1.94 mm- 1.54 mm). The whole mPFC (all subregions pooled together) showed an increase in activated, pERK⁺ neurons following novel taste consumption compared to familiar taste (unpaired t-test: t=5.359, DF=14, p=0.0001; *Figure 3b*). Analysis of the different layers yielded significant effects both in the outer (t=5.835, DF=14, p<0.0001; *Figure 3c*) and inner (t=4.512, DF=14, p=0.0005; *Figure 3d*) layers of the mPFC. The changes in the total mPFC are derived from both the Cg1 (t=3.111, DF=14, p=0.0077; *Figure 3e*), and the PrL (t=4.306, DF=14, p=0.007; *Figure 3f*), but not the InfrL (t=0.6712, DF=14, p=0.513; *Figure 3g*). A significant increase in activated mPFC-to-aIC projecting neurons following

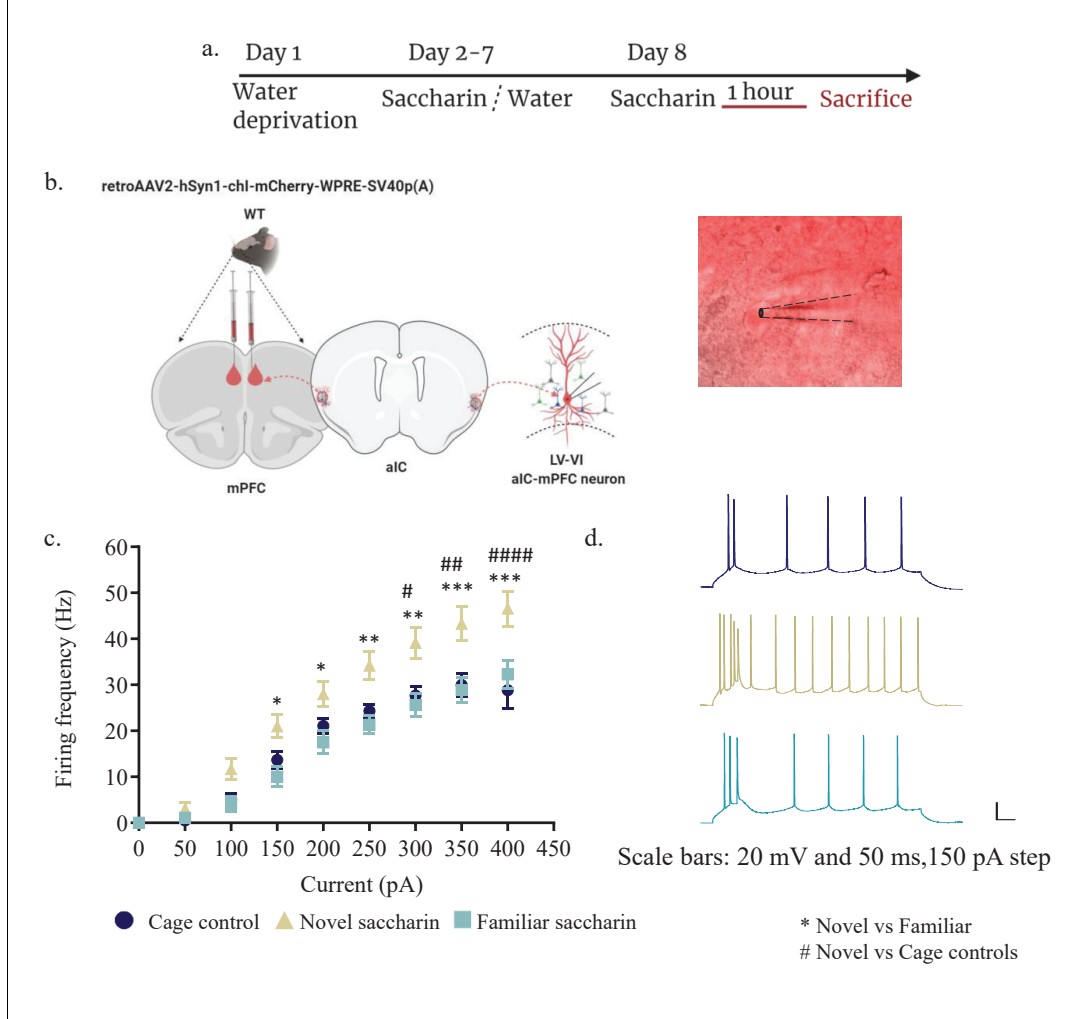

**Figure 2.** Novel taste exposure increases excitability of mPFC-projecting neurons of the inner insular layers. (a) Schematic representation of novel or familiar taste learning. Animals were water restricted in the first day and administered with saccharin (familiar saccharin group) or water (novel saccharin group) in the following 6 days. Mice were given novel or familiar saccharin and sacrificed 1 hr later. Intrinsic properties were measured in the different layers of the aIC. (b) Whole-cell current-clamp recordings of aIC neurons projecting to mPFC (red). (c, d) The dependence of firing rate on current step magnitude is significantly higher in layers V/VI insular neurons projecting to mPFC following novel compared to familiar taste or cage controls (two-way repeated measurements ANOVA, n=13–14 cells per group; p=0.0009; F (2, 37) = 8.465). Error bars represent SEM; *p<0.05, **p< 0.01, ***p<0.001, ****p<0.0001.

The online version of this article includes the following source data and figure supplement(s) for figure 2:

**Source data 1.** Novel taste exposure increases excitability of mPFC projecting neurons of the inner insular layers.

**Figure supplement 1.** Layer II/III of mPFC projections of the aIC displays similar intrinsic properties after novel and familiar taste exposure.

**Figure supplement 1—source data 1.** Layer II/III of mPFC projections of the aIC displays similar intrinsic properties after novel and familiar taste exposure.

**Figure supplement 2.** Novel taste exposure increases excitability of mPFC-projecting neurons of the inner insular layers.

**Figure supplement 2—source data 1.** Novel taste exposure increases excitability of mPFC-projecting neurons of the inner insular layers.

novel compared to familiar taste was also identified (t=9.549, DF=14, p<0.0001; *Figure 3j*). The increase in activated aIC-projecting neurons of the mPFC is derived from both outer (t=5.504, DF=14, p<0.0001; *Figure 3k*), and inner layers (t=9.923, DF=14, p<0.0001; *Figure 3i*) of the mPFC. Additionally, all mPFC subregions exhibited increased activation of IC-projecting neurons (Cg, t=4.126, DF=14, p=0.001; PrL, t=9.705, DF=14, p<0.0001; InfrL, t=5.141, DF=14, p=0.0001; *Figure 3m–o*). Analysis of number of rAAV⁺ neurons for the different subregions/layers of the mPFC showed no significant difference between novel/familiar saccharin groups (*Figure 1—figure*

**Table 1.** Summary of intrinsic properties of aIC-to-mPFC projecting neurons following novel or familiar taste experience.

| Groups | RMP (mV) | mAHP (mV) | Input resistance (MΩ) | Sag ratio (%) | Time constants (ms) | AP thresh (mV) | AP Amp (mV) | AP half-width (ms) | Rheobase (pA) |
|---|---|---|---|---|---|---|---|---|---|
| Layer 2/3 IC-mPFC (F.Sacc) | −72.26 ± 2.02 (10) | −4.223 ± 0.8426 (10) | 128.9 ± 15.02 (10) | 6.96 ±1.633 (10) | 21.89 ± 2.855 (10) | −30.2 ± 1.845 (10) | 52.61 ± 3.211 (10) | 0.696 ± 0.04483 (10) | 118.2 ± 31.24 (10) |
| Layer 2/3 IC-mPFC (Novel) | −71.65 ± 1.504 (10) | −3.509 ± 0.8405 (10) | 128.6 ± 9.645 (10) | 5.631 ±1.298 (10) | 19.63 ± 2.623 (10) | −33.92 ± 1.109 (10) | 55.06 ± 2.263 (10) | 0.703 ± 0.03798 (10) | 119.9 ± 16.23 (10) |
| Layer 5/6 IC-mPFC (F.Sacc) | −62.18 ± 1.368 (13) * | −4.14 ± 0.6927 (13) | 106.9 ± 11.32 (13) * | 12.29 ± 1.993 (13) | 14.27 ± 1.842 (13) | −33.08 ± 2.421 (13) | 56.4 ± 2.135 (13) | 0.6931 ± 0.05582 (13) | 102.7 ± 13.68 (13) * |
| Layer 5/6 IC-mPFC (Novel) | −67.21 ± 1.215 (14) | −5.163 ± 0.916 (14) | 150.1 ± 12.94 (14) | 8.349 ± 2.134 (14) | 18.4 ± 2.049 (14) | −35 ± 1.499 (14) | 50.53 ± 2.242 (14) | 0.6136 ± 0.03875 (14) | 68.86 ± 8.601 (14) |
| Layer 5/6 IC-mPFC (Cage controls) | −68.37 ± 1.177 (13) ++ | −3.359 ± 0.9374 (13) | 128.5 ± 10.54 (13) | 14.11 ± 2.752 (13) | 18.74 ± 2.406 (13) | −33.17 ± 1.884 (13) | 53.82 ± 2.298 (13) | 0.8308 ± 0.05237 (13) $ | 81.62 ± 10.71 (13) |

*=p<0.05, aIC-to-mPFC L5/6 novel vs familiar saccharin. ++ = p<0.01 aIC-to-mPFC L5/6 cage control vs familiar saccharin. $=p<0.05 aIC-to-mPFC L5/6 novel saccharin vs cage controls.

supplement 1c,d). These results show that the reciprocal aIC-to-mPFC circuit is activated following novel taste consumption.

## Activation of aIC-to-mPFC projecting neurons is necessary for both novel taste neophobic responses and memory formation

In many correlative experimental set-ups, it is difficult to dissociate between the cellular activity needed for expressing the behavior from that which is needed to acquire the information for a learning process. Based on the correlation yielded above (*Figures 1–3*), between novel taste consumption and the reciprocal activation of aIC to/from mPFC, we sought to identify whether activation of the aIC-to-mPFC part of the circuit is necessary for novel taste neophobic responses and/or familiarization learning. We therefore injected mice with retrograde-Cre virus in the mPFC, and Cre-dependent inhibitory DREADDs (hM4Di) in the aIC (Materials and methods, *Figure 4a,b*). Although we used a low concentration of CNO, in order to account for any nonspecific effects of CNO administration (*Gogolla, 2017*), we injected another group of animals with AAV constructs at the aIC and mPFC, resulting in the expression of mCherry, without DREADD receptors in aIC-to-mPFC projections (see Materials and methods). Animals expressing hM4Di in the aIC-to-mPFC projecting neurons were administered either CNO (n=11) or saline (n=10), 1 hr prior to a choice test of novel saccharin (first exposure) and water (*Figure 4c*). Choice tests between water and saccharin were then carried out over the next 4 days. We found that inhibition of aIC-to-mPFC projecting neurons not only impairs the neophobic response in the first day (unpaired t-test; p=0.0095, t=2.883, DF=19; *Figure 4d*), but also causes a measurable increase in the neophobic response on the next day (two-way ANOVA: F(1,175)=7.558, p=0.0075; *Figure 4e* and *Figure 4—figure supplement 1a*). In order to control for possible effects of CNO, additional group of mice that were injected with control virus that does not express hM4Di in aIC-to-mPFC projections, were treated with CNO (n=7) 1 hr prior to the same experimental design. Aversion toward the novel taste and its attenuation were not different from the saline treated group (*Figure 4—figure supplement 1a*), which excludes a possible effect of CNO itself.

Our results indicate a necessity for the circuit in denoting the taste as novel, although it does not rule out its effect on other facets such as taste recognition, retrieval or aversion.

In order to test recognition and retrieval of a familiar taste, we inhibited the circuit on the seventh day, 1 hr prior to a choice test between saccharin and water, once saccharin neophobia has been attenuated and the taste is no longer novel. We observed no significant difference between the treatment (n=5) and control (n=6) groups (unpaired t-test: t=0.2364, DF=9, p=0.8184; *Figure 4—figure supplement 1b,c*). In order to test whether inhibiting aIC-to-mPFC projections affects taste aversion retrieval, we trained animals in CTA and inhibited the circuit one hour before the retrieval test (see Materials and methods). We observed no significant difference between animals treated with

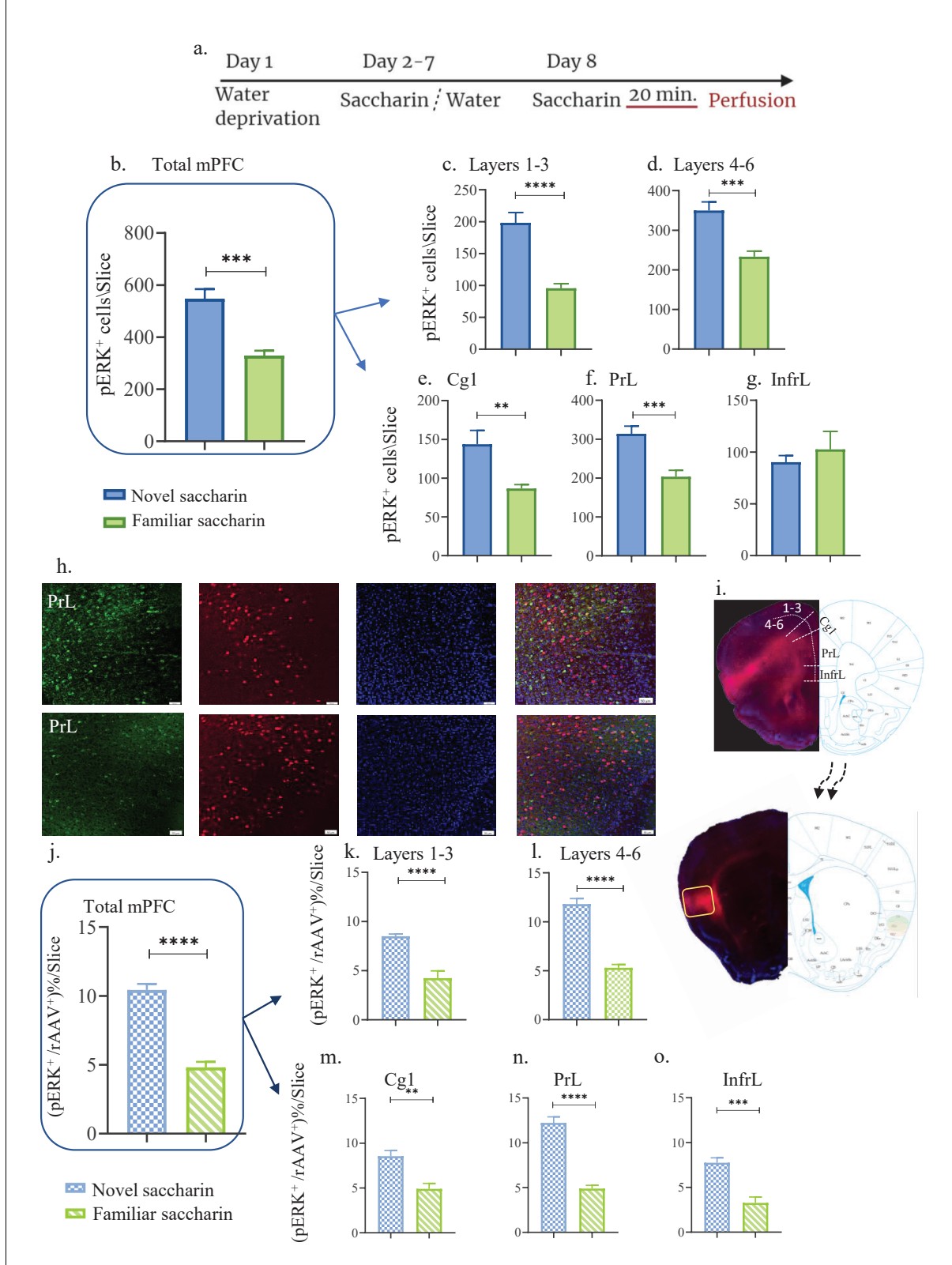

**Figure 3.** Novel taste experience increases number of pERK+ cells in aIC-projecting neurons of the mPFC. (a) Schematic representation of novel or familiar taste learning; animals were water restricted in the first day and administered with saccharin (familiar saccharin group) or water (novel saccharin group) in the following 6 days. In the last day, mice were presented with 1 ml of Saccharin 20 min prior to perfusion.number of pERK+ neurons was quantified in the different subregions and layers of the mPFC. (b) Number of pERK+ neurons of the mPFC was significantly increased following novel

*Figure 3 continued on next page*

Figure 3 continued

(548.1±36.24) compared to familiar (329±18.89) saccharin consumption. (c) Number of pERK$^+$ neurons of the outer layers of the mPFC was significantly increased following novel (198±16.15) compared to familiar (95.75±6.920) saccharin consumption. (d) Number of pERK$^+$ neurons of the inner layers of the mPFC was significantly increased following novel (349±21.67) in comparison to familiar (233.4±14.040) saccharin consumption. (e) Number of pERK$^+$ neurons in the Cg1 was significantly higher following novel (143±17.71) compared to familiar (86.63±4.988) saccharin consumption. (f) Number of pERK$^+$ neurons in the PrL was significantly higher following novel (314±19.68) in comparison to familiar (203.6±16.44) saccharin consumption. (g) Number of pERK$^+$ neurons in the InfrL was similar following novel (90.23±6.423) and familiar (102.6±17.32) saccharin consumption. (h) Representative coronal mPFC sections immunostained for pERK (Green) and DAPI (blue) from mice injected with rAAV at the IC (Red) following novel (upper) and familiar (lower) saccharin. Scale bar, 50 µm, 20x. (i) Stereotaxic injection of rAAV-mCherry construct (red) at the IC and its subsequent labeling in the mPFC. Representative schematic overlays of the Cre-dependent expression of the chemogenetic receptors using the rAAV systems is shown, demonstrating the expressionto be restricted in the aIC and mPFC. Percentage of double labeled (pERK$^+$,rAAV$^+$) neurons was calculated as a percentage of all rAAV$^+$ neurons. (j) Percentage of double-labeled neurons of the mPFC was significantly increased following novel (10.44 ± 0.434%) compared to familiar (4.819 ± 0.397%) saccharin consumption. (k) Percentage of double-labeled neurons of the outer layers of the mPFC was significantly increased following novel (8.49 ± 0.230%) compared to familiar (4.23 ± 0.739%) saccharin consumption. (l) Percentage of double-labeled neurons of the inner layers of the mPFC was significantly increased following novel (11.81 ± 0.561%) compared to familiar (5.296 ± 0.340%) saccharin consumption. (m) Percentage of double-labeled neurons of the Cg1 was significantly increased following novel (8.543 ± 0.633%) compared to familiar (4.897±0.617) saccharin consumption. (n) Percentage of double-labeled neurons of the Prl was significantly increased following novel (12.23 ± 0.665%) in comparison to familiar (4.893 ± 0.358%) saccharin consumption. (o) Percentage of double-labeled neurons of the InfrL was significantly increased following novel (7.744 ± 0.554%) in comparison to familiar (3.261 ± 0.673%) saccharin consumption. Data are shown as mean ± SEM. *p<0.05, **p< 0.01, ***p<0.001, ****p<0.0001.

The online version of this article includes the following source data for figure 3:

**Source data 1.** Novel taste experience increases number of pERK cells in aIC-projecting neurons of the mPFC.

CNO (n=11) or saline (n=11; unpaired t-test: t=0.4114, DF=20, p=0.6852; *Figure 4—figure supplement 1d,e*).

Furthermore, we inhibited the aIC-to-mPFC part of the circuit prior to a choice test between innately aversive quinine and water (CNO=7, Saline=7), and found no significant difference in aversion between the two groups (unpaired t-test: t=1.102, DF=12, p=0.2919; *Figure 4—figure supplement 1f,g*). These results therefore indicate that the aIC-to-mPFC part of the circuit is not necessary for the retrieval of a familiar taste, taste aversion or taste recognition, but is vital for both the initial neophobic response, as well as incidental memory formation.

We have previously shown that activity within aIC-to-BLA projecting neurons is not necessary for attenuation of neophobia (*Kayyal et al., 2019*). However, since activity within the BLA is necessary for the expression of a neophobic response (*Shinohara and Yasoshima, 2019*; *Lin et al., 2018*) and in order to better understand the broader circuit involved in the expression of a neophobic reaction, we sought to inhibit aIC neurons projecting to the BLA during the first encounter with novel saccharin. Mice were injected with retrograde-Cre AAV at the BLA, and Cre-dependent inhibitory DREADDs at the IC (see Materials and methods, *Figure 5a–b*). A month later, mice received an i.p. injection of CNO (n=8) or saline (n=6) one hour prior to a choice test of novel saccharin and water (*Figure 5c*). Inhibition of aIC-to-BLA projecting neurons caused a reduction in neophobia (unpaired t-test; p=0.007, t=3.243, DF=12) compared to the control group (*Figure 5d*). However, the attenuation curve did not differ between the two treatments in accordance with our previous report (two way ANOVA: F(1,43) = 0.07789, p=0.7815; *Figure 5e*), indicating no change in novelty learning. Together, the results show that aIC-to-mPFC projections are necessary both for learning and expressing taste neophobia, while aIC-to-BLA projections are necessary only for the expression of the neophobic response, but not for its attenuation.

## Activity of mPFC-to-aIC projecting neurons is necessary for novel taste neophobic response but not memory formation

To better understand the interplay of information between the mPFC and IC and the role the mPFC plays in novelty behavior, we sought to inhibit the directionally opposite element of the pathway, the mPFC-to-aIC projection. Mice were therefore injected with rAAV-Cre at the aIC, and Cre-dependent inhibitory DREADDs at the mPFC (see materials and methods, *Figure 6a–b*). We then injected mice with CNO (n=8) or Saline (n=5) one hour prior to a choice test between novel saccharin (first exposure) and water (*Figure 6c*). Inhibition of the mPFC-to-aIC circuit caused a reduction in aversion index during the first choice test (unpaired t-test: t=2.995, DF=11, p=0.0131; *Figure 6d*). However,

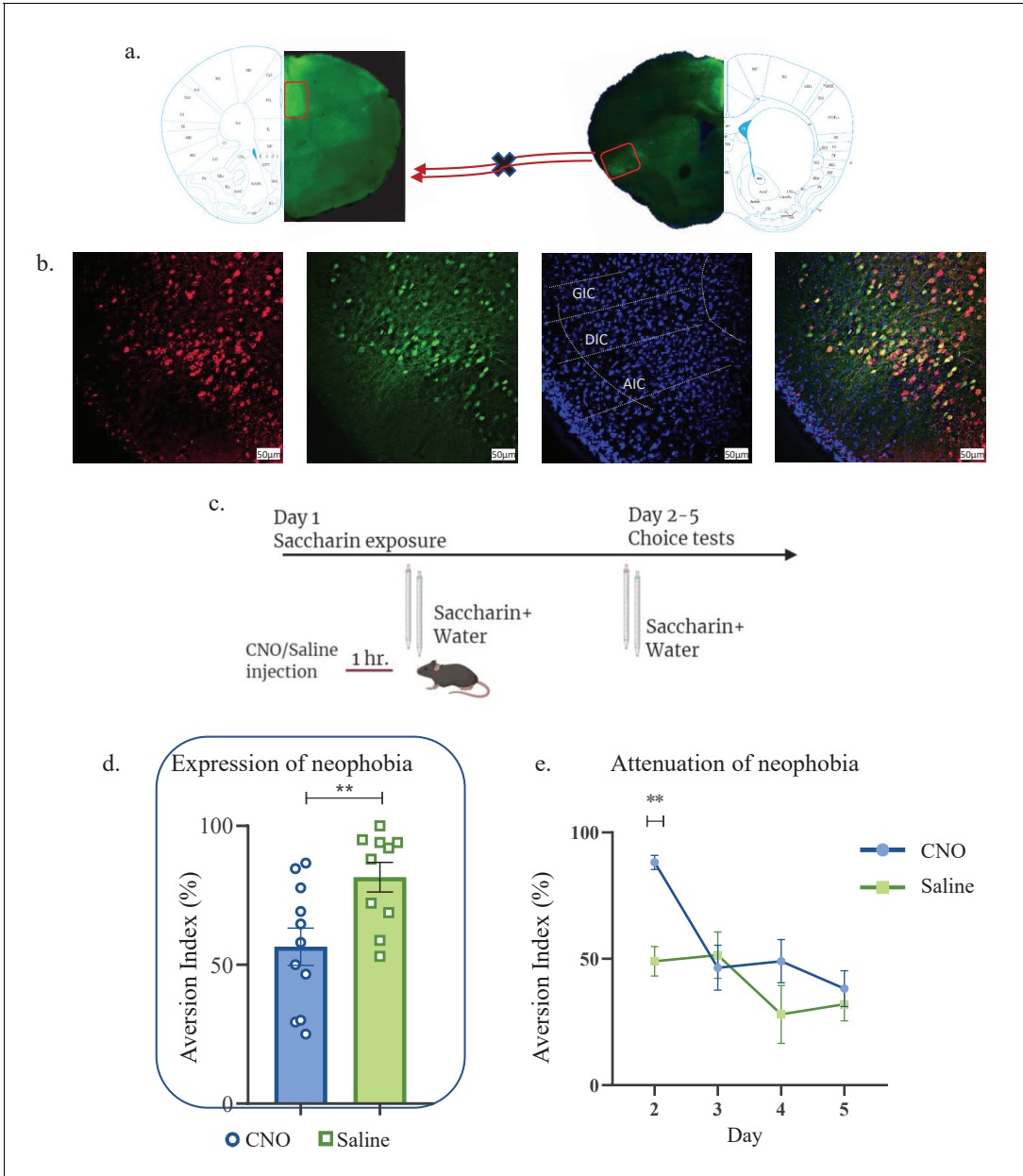

**Figure 4.** aIC-to-mPFC projecting neurons are necessary for both novel taste neophobic response and learning. WT male mice were injected with retrograde-Cre AAV at the mPFC (green) and the Cre-dependent DREADDs at the IC (red), resulting in expression of inhibitory hMD4Gi in aIC-to-mPFC neurons. (a) Representative images of retrograde-Cre AAV injection in the mPFC and the consequent labeling in the IC (red). (b) Representative images of neurons targeted for inhibition in the IC: Cre-dependent expression of the chemogenetic receptors (red), retrograde-Cre AAV (green) and DAPI (blue). (c) Schematic representation of behavioral test conducted. Mice injected with IC-to-mPC inhibitory DREADDs received intraperitoneal injection of CNO or Saline 1 hr prior to novel saccharin (first exposure) and water choice test, while aversionwas assessed. In the following days, mice were given a choice test without intervention. (d) CNO-injected mice exhibited significantly lower aversion (51.21 ± 8.255%) levels than do Saline-injected controls (78.36 ± 4.33%) in the first choice day (expression of neophobia). (e) CNO-injected mice were significantly more averse to saccharin (87.45 ± 4.336%, compared to saline (54.57 ± 5.708%)) injected controls in the second day. CNO-injected and saline-injected mice showed similar aversion in the third (CNO: 60.70 ± 8.599%, Saline: 54.93 ± 12.68%) fourth (CNO: 51.13 ± 7.804%, Saline: 27.29 ± 14.34%) and fifth (CNO: 45.77 ± 7.852%, Saline: 29.63 ± 6.974%) day of choice tests. Data are shown as mean ± SEM. *p<0.05, **p< 0.01, ***p<0.001, ****p<0.0001.

The online version of this article includes the following source data and figure supplement(s) for figure 4:

**Source data 1.** aIC-to-mPFC projecting neurons are necessary for both novel taste neophobic response and learning.
**Figure supplement 1.** aIC-to-mPFC inhibition does not affect taste recognition.
**Figure supplement 1—source data 1.** aIC-to-mPFC inhibition does not affect taste recognition.

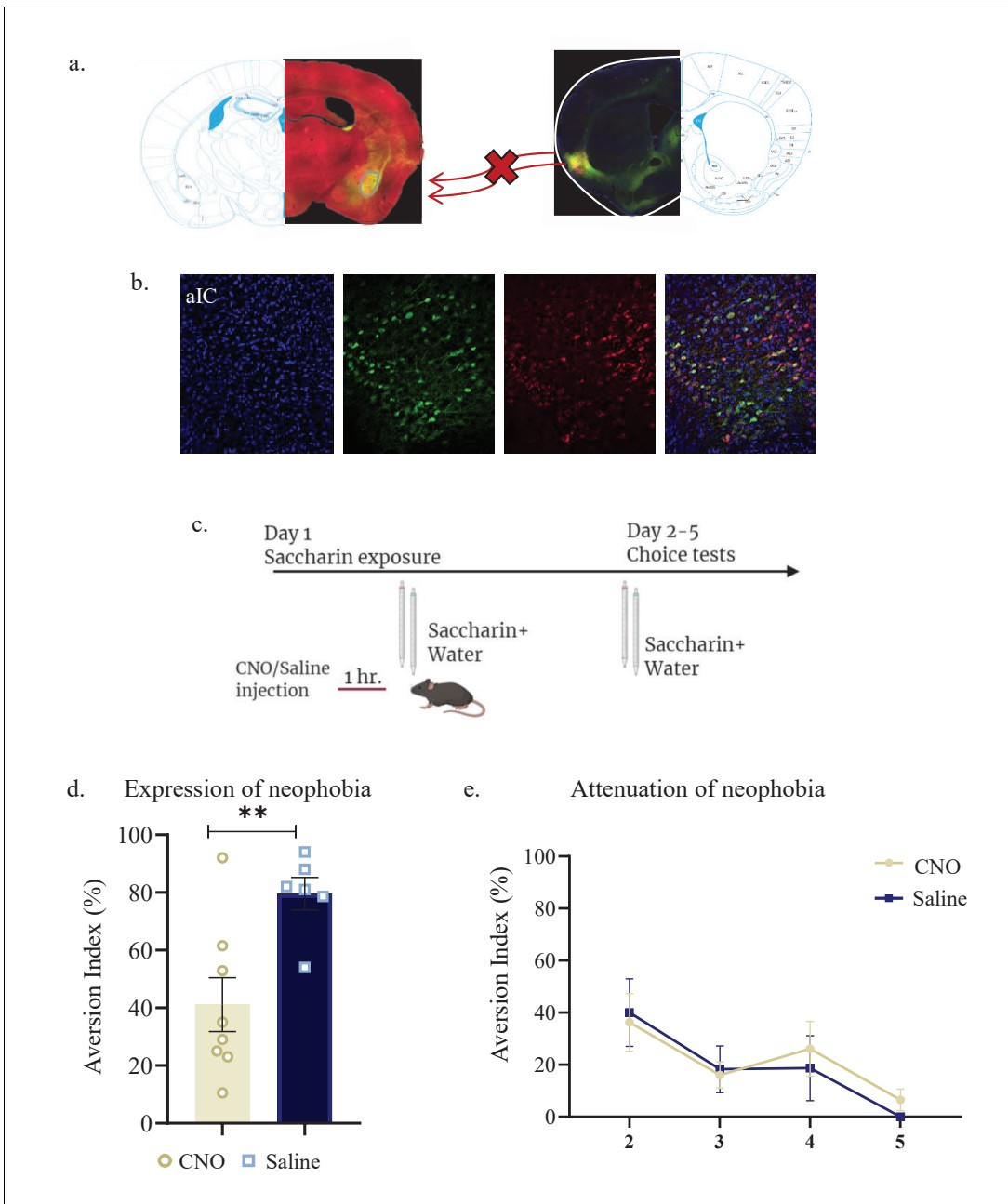

**Figure 5.** Activity within aIC-to-BLA projecting neurons is necessary for novel taste response but not for its attenuation. WT male mice were injected with retrograde-Cre AAV at the BLA (green) and Cre-dependent DREADDs at the IC (red), resulting in expression of inhibitory hMD4Gi in aIC-to-BLA projecting neurons. (a) Representative images of retrograde-Cre AAV injection in the BLA (green) and the consequent labeling in the IC. (b) Representative images of IC-to-BLA projecting neurons that are targeted for inhibition using Cre-dependent expression of the chemogenetic receptors (red), retrograde-Cre AAV (green) and DAPI (blue). (c) Schematic representation of behavioral test conducted. Mice injected with IC-to-BLA inhibitory DREADDs received intraperitoneal injection of CNO or Saline 1 hr prior to novel saccharin (first exposure) and water choice test, while aversion was assessed. In the following days, mice were given a choice test without intervention. (d) CNO-injected mice exhibited lower aversion (41.11 ± 9.309%) levels than do Saline-injected controls (79.60 ± 5.604%) in the first choice day (unpaired t-test: p=0.007, t=3.243, DF=12). (e) CNO-injected and saline-injected mice showed similar aversion on the second (CNO: 36.26 ± 11.04%, Saline: 40.0 ± 12.98%), third (CNO: 16.06 ± 5.01%, Saline: 18.25 ± 8.920%), fourth (CNO: 26.13 ± 10.52%, Saline: 18.67 ± 12.48%), and fifth (CNO: 6.50 ± 4.153%, Saline: 0.00 ± 0.00%) day of expression of neophobia. Data are shown as mean ± SEM. *p<0.05, **p< 0.01, ***p<0.001, ****p<0.0001.
The online version of this article includes the following source data for figure 5:

**Source data 1.** Activity within aIC-to-BLA projecting neurons is necessary for novel taste response but not for its attenuation.

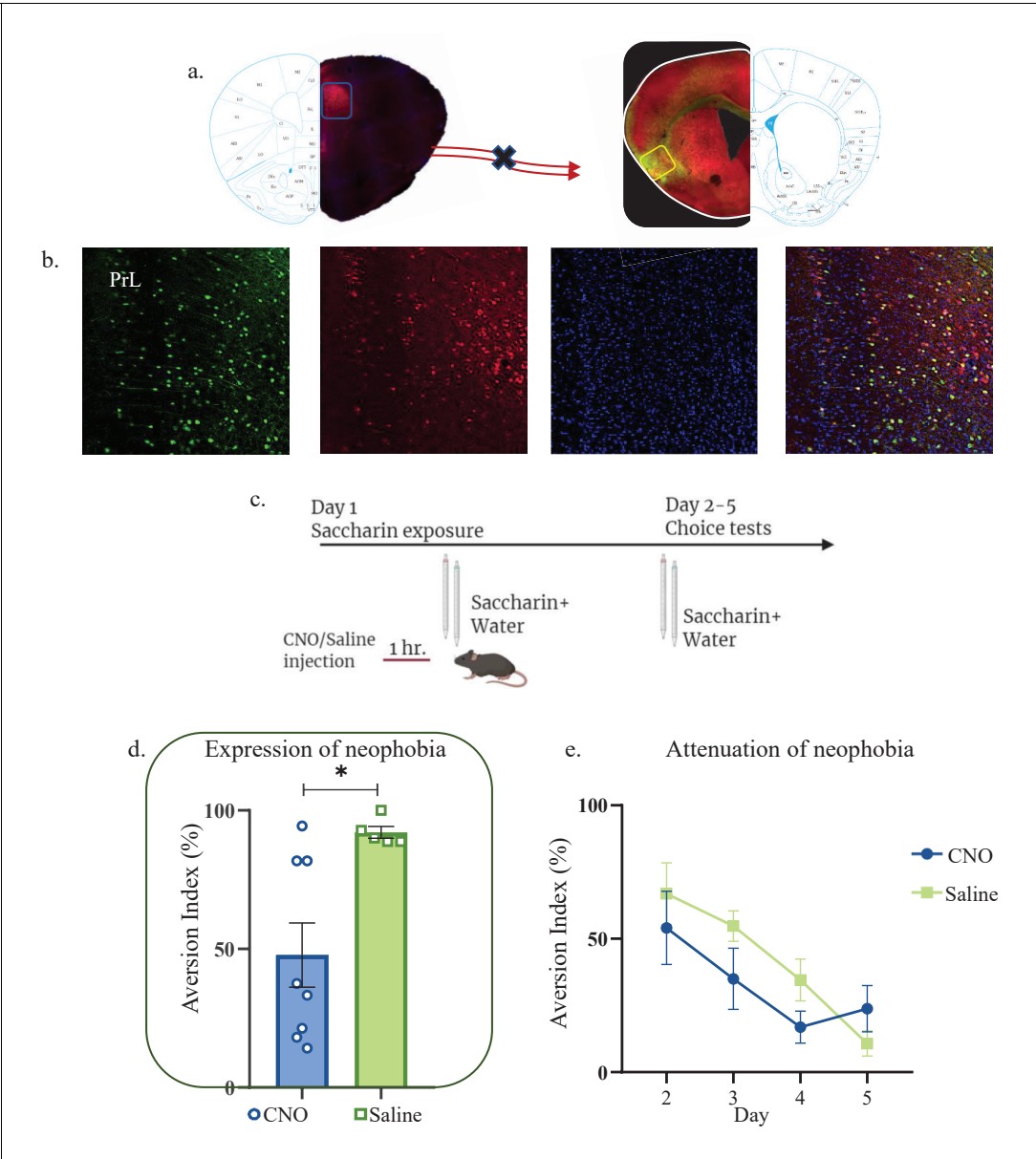

**Figure 6.** Activity of mPFC-to-aIC projecting neurons is necessary for novel taste neophobic reaction but not for the learning. WT male mice were injected with Cre-dependent DREADDs at the mPFC and rAAV-Cre at the IC, resulting in expression of inhibitory hMD4Gi in mPFC-to-aIC neurons. (a) Representative images of retrograde-Cre AAV injection in the IC (red) and the consequent labeling in the mPFC. (b) Representative images of mPFC-to-aIC projecting neurons that are targeted for inhibition: Cre-dependent expression of the chemogenetic receptors (red), retrograde-Cre AAV (green) and DAPI (blue). (c) Schematic representation of behavioral test conducted. Mice injected with mPFC-to-IC inhibitory DREADDs received intraperitoneal injection of CNO or Saline 1 hr prior to novel saccharin (first exposure) and water choice test, while aversion was assessed. In the following days, mice were given a choice test without intervention. (d) CNO-injected mice (47.81 ± 11.58%) exhibited significantly lower aversion levels than do saline-injected controls (92.08 ± 2.111%) in the first choice day (expression of neophobia). (e) CNO-injected and saline-injected mice showed similar aversion on the second (CNO: 54.06 ± 13.71%, Saline: 66.84 ± 11.60%), third (CNO: 34.98 ± 11.44%, Saline: 54.76 ± 5.693%), fourth (CNO: 16.85 ± 5.986%, Saline: 34.54 ± 7.856%), and fifth (CNO: 23.80 ± 8.681%, Saline: 10.74 ± 4.712%) day of attenuation of neophobia. Data are shown as mean ± SEM. *p<0.05, **p< 0.01, ***p<0.001, ****p<0.0001.

The online version of this article includes the following source data and figure supplement(s) for figure 6:

**Source data 1.** Activity of mPFC-to-aIC projecting neurons is necessary for novel taste neophobic reaction but not for the learning.

**Figure supplement 1.** mPFC-to-aIC projecting neurons are necessary for novel taste response but not for familiar taste taste retrieval.

**Figure supplement 1—source data 1.** mPFC-to-aIC projecting neurons are necessary for novel taste response but not for familiar taste retrieval.

**Figure supplement 2.** Inhibition of aIC-to-mPFC and mPFC-to-aIC projecting neurons does not affect anxiety-related behavior.

**Figure supplement 2—source data 1.** Inhibition of aIC-to-mPFC and mPFC-to-aIC projecting neurons does not affect anxiety-related behavior.

it did not affect attenuation of neophobia in the following days (two way ANOVA: F(1,44) = 1.608, p=0.2114; *Figure 6e*, *Figure 6—figure supplement 1a*). In order to test whether this was due to an effect on taste identification, we performed another experiment in which we injected animals with CNO (n=4) or saline (n=5) prior to a choice test, conducted after mice were familiarized with saccharin (six exposures; *Figure 6—figure supplement 1b*). Inhibition of mPFC-to-aIC projecting neurons did not affect aversion after taste familiarization (*Figure 6—figure supplement 1c*; unpaired t-test: t=0.3018, DF=7, p=0.7716). This indicates that the activation of mPFC-to-aIC projecting neurons is necessary for the neophobic response to a novel taste, but not for the formation of a memory trace, as is additionally seen in the reciprocal aIC-to-mPFC part of the pathway.

To further investigate whether aIC-mPFC inhibition affects anxiety levels, that may be associated with and/or affecting novel taste experiences, we performed an open-field test (*Seibenhener and Wooten, 2015*). Groups expressing hM4Di in aIC-to-mPFC (n=8) or mPFC-to-aIC projecting neurons (n=9), as well as their respective control virus injected groups (n=8), were all treated with i.p. injections of CNO. One hour later, mice were placed in an open arena (*Figure 6—figure supplement 2a*). Time spent in the center of the arena and frequency of crossing the center were analyzed. No significant differences was found in both parameters, indicating no effect on anxiety as a consequence of inhibition of aIC-mPFC reciprocal connectivity (*Figure 6—figure supplement 2b,c*). Aiming to clarify further whether the circuit manipulations affect the detection of the novel taste or anxiety associated with presentation of this taste, we performed another set of experiments, in which we tested the latency toward a novel saccharin and water presentation. Groups expressing hM4Di in aIC-to-mPFC (n=8) or mPFC-to-aIC projecting neurons (n=9) as well as their respective control virus injected groups (n=9) were all injected i.p with CNO. Mice were then presented with a pipette of novel saccharin and time to approach the pipette was recorded. Following 5 s of consumption, the saccharin pipette was removed and 10 s later a water pipette was presented. This routine of saccharin/water presentation was performed twice, consecutively (*Figure 6—figure supplement 2d*). No significant difference was found between the groups in the time it took the mice to start drinking, either during the first or second presentation of the novel saccharin or water (*Figure 6—figure supplement 2e*), further indicating no effect on anxiety levels caused by our intervention.

## Discussion

Detecting salient, new information from the environment is a crucial aspect of human and animal behavior. The detection and proper reaction to a novel stimulus, and the ability to form a stable memory of it, are two distinct facets of novelty behavior. The molecular and cellular mechanisms underlying these centrally important processes, which are critical for animal survival, have been the subject of much research (*Mishkin and Murray, 1994*; *Schomaker and Meeter, 2015*). As a result, numerous studies have explored the role of distinct brain areas and molecular or cellular processes within them that are crucial for salience and novelty detection and learning (*Peters et al., 2016*; *van Kesteren et al., 2012*). Although studies conducted in humans show a correlation between several brain structures and novelty detection, little is yet known of the detailed novelty network and directionality in the brain, resulting from causative research (*Li et al., 2017*). It is therefore important to understand how functionally relevant, distal brain regions act as a circuit to define salience and encode unfamiliar sensory information within a wider salience system.

In the present study, we causally demonstrate that reciprocal activity between the aIC and mPFC, or the BLA, conveys novelty related gustatory information, in one such novelty circuit. Specifically, we show that general neuronal activation in the aIC, as denoted by pERK (*Thiels and Klann, 2001*), is increased following novel taste experience. This is in line with previous studies, showing that ERK phosphorylation in the aIC allows the formation of a memory trace of the safe taste, and is correlated with taste novelty (*Sweatt, 2001*; *Gutkind, 1998*; *Kelleher et al., 2004*; *Thomas and Huganir, 2004*; *Yiannakas and Rosenblum, 2017*). Importantly, our results indicate pERK is increased specifically in the AIC and the DIC subregions of the aIC, which have previously been implicated in processing chemosensory, somatosensory visceral and limbic functions (*Jones et al., 2006*; *Katz et al., 2001*; *Yokota et al., 2011*), emphasizing their relevance in novel taste learning.

Extensive reciprocal connections exist between the IC and the mPFC, that are thought to be an essential component in salience processing both in humans and rodents (*Uddin, 2015*). Here too,

our results show a correlation between the number of activated, pERK$^+$ neurons within reciprocal aIC-mPFC projections following novel taste experience, suggesting plasticity related changes in aIC-mPFC neurons are initiated following taste learning (*Impey et al., 1999*; *Philips et al., 2013*; *Purcell et al., 2003*; *Rosenblum et al., 2002*). Interestingly, in the aIC this was particularly apparent in the AIC, while in the mPFC the spread of pERK$^+$ neurons was more generalized, being detected across the PrL, InfrL and Cg. The projections of the aIC to the mPFC mainly originate from the AIC (compared to the DIC or GIC), which could be a possible explanation for the specific subregion-effect.

Recent reports indicate the capability of PrL neurons to form ensemble codes for novel stimulus associations within minutes (*Takehara-Nishiuchi et al., 2020*). This ability seems to underlie the necessity of the mPFC to rapidly detect and selectively encode novel experiences, although our data suggest a role for all mPFC subregions and layers in novel taste processing and learning. Whether and how mPFC subregions, as well as the emerging local circuitry of the aIC are involved in novel taste learning via cell- and subregion-specific mechanisms, on a molecular and cellular level across different phases of learning, will be the subject of future research.

We found that ERK activation within aIC-to-PFC projections correlated with taste novelty, and given ERK involvement in maintaining long-term memory-relevant excitability changes (*Cohen-Matsliah et al., 2007*), we assessed the intrinsic properties of these projections subsequent to novel taste consumption. In line with the pERK results, we found that aIC-to-mPFC neurons display distinct electrophysiological properties following novel taste consumption. This is in agreement with previous studies in other modalities, showing that novel stimuli and learning of associative experience differentially affect intrinsic properties of neurons in cortical and subcortical regions (*Kaczorowski et al., 2012*; *Sehgal et al., 2013*; *Sehgal et al., 2014*; *Song et al., 2015*; *Song et al., 2015*). The increased excitability of aIC-to-mPFC neurons following novel taste experience was observed only in the deep layer of the aIC. Limbic and gustatory information converge in the deep layers of the entire IC, and the main output source of the IC to other cortical and subcortical regions lies within these layers (*Kobayashi, 2011*; *Krushel and van Der Kooy, 1988*). In addition, the percentage of taste-coding neurons is higher in the deep layers of the aIC in comparison with the superficial layers (*Dikecligil et al., 2020*). Our results further strengthen the involvement of the deep layers of the aIC in processing and transmitting taste-related information to other cortical regions, suggesting that these specific molecular and intrinsic changes of aIC-to-mPFC neurons of the deep layers of the aIC allow the formation of a familiar and safe taste memory trace.

Importantly, even though the correlated increase in pERK and in excitability of aIC-mPFC projections following a novel taste demonstrate the involvement of this circuit in novel taste processing, it does not differentiate between the two components of novel taste behavior: novel taste response, or novel taste learning. We thus evaluated if indeed there is a functional aIC-mPFC circuit, by using chemogenetic inhibition to causally examine the reciprocal role of both components, the aIC and the mPFC, in novel taste expression or learning. We found that inhibition of aIC-to-mPFC projecting neurons during novel taste exposure resulted in impaired neophobic responses and novel taste learning. In contrast, inhibiting mPFC-to-aIC projections impaired the expression of neophobia, but not taste memory formation.

This indicates that the mPFC plays a major role in novelty gating, while the detection of novelty and storage of familiar taste memories are both mediated by the aIC. We show that aIC-to-mPFC projections need to be activated during novel taste exposure in order to form a safe taste memory trace, while aIC activation from the mPFC is necessary in order to evoke appropriate behavioral responses to novelty. We thus causally demonstrate that specific connectivity within the aIC and the mPFC allows the identification, the learning and reaction to novel taste stimuli with the memory being formed in the aIC. However, this does not exclude the involvement of other additional stimuli or other circuits that were not the focus of the current study.

The entire IC is an integrative hub for saliency processing. We recently demonstrated that the reciprocal connectivity with the BLA underlies aversive valence processing (*Kayyal et al., 2019*; *Lavi et al., 2018*). The BLA is an essential structure for encoding salient stimuli (*Fontanini and Katz, 2009*) and reward learning (*Fontanini et al., 2009*), and its signals to the aIC transmit palatability related information (*Piette et al., 2012*), while facilitating the acquisition and recall of sensory information related to body states, as well as the processing of valence (*Gogolla, 2017*; *Haley and Maffei, 2018*; *Tye, 2018*). Although we have previously shown that activity within aIC-to-BLA projecting

neurons is not necessary for novel taste learning, we did not test if it is necessary for the expression of taste neophobia. Here, we show that activity within aIC-to-BLA projecting neurons is essential for the expression of neophobic responses. This goes along with previous data showing that inactivation of the BLA impairs the neophobic response but does not affect the process of attenuation of neophobia, and therefore learning (*Lin et al., 2018*; *Shinohara and Yasoshima, 2019*).

Together, our findings suggest that aIC connectivity to different brain areas differentially conveys the novelty state (aIC-to-mPFC/BLA), while following learning-induced plasticity, a now modified circuit state underlies the encoding of the familiarized taste (aIC-to-mPFC but not to BLA) (*Figure 7*). A functional reciprocal connectivity exists between the BLA and the mPFC (*Yizhar and Klavir, 2018*),

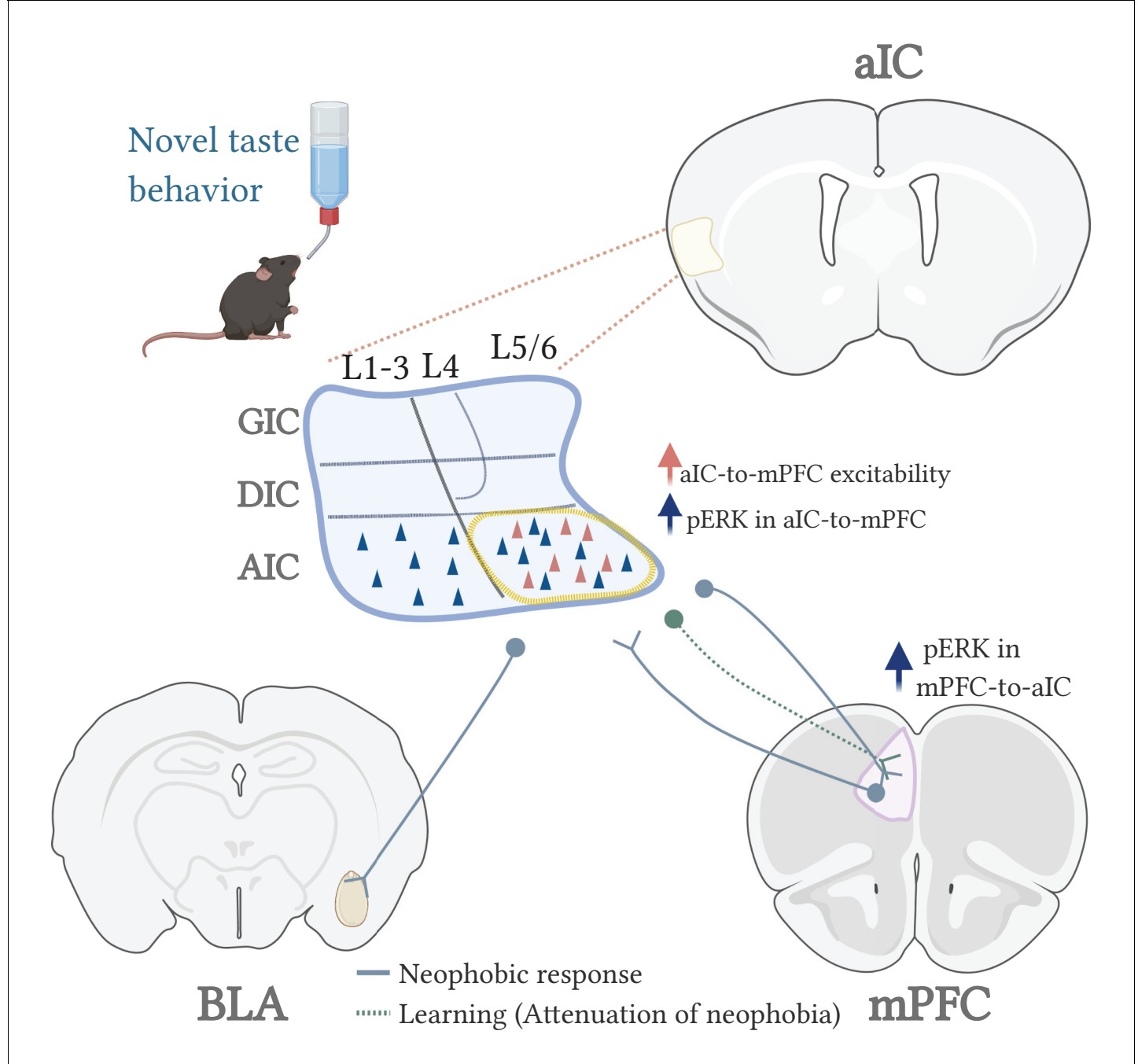

**Figure 7.** A model for the circuits involved in novel taste expression and learning.

that is of great importance in relation to drinking behavior and water intake (*Zhao et al., 2020*). In addition, activity within BLA-mPFC projections is necessary for valence-specific behaviors (*Yizhar and Klavir, 2018*) including innate fear responses (*Jhang et al., 2018*). Hence, our results do not exclude the possibility of mPFC-BLA activity during novel taste experience and/or learning, but rather suggest a triage of IC, BLA and mPFC participation in novel taste behavior.

Detecting, responding and remembering salient information is a vital property of animal behavior. We show here, for the first time, that the interplay between the aIC and the mPFC in the mammalian brain grasps within it novelty related data, as part of a salience network. Both aIC-mPFC and aIC-to-BLA reciprocal connections are necessary for exhibiting a neophobic response; however, the aIC-mPFC is necessary also to form a memory for a familiar safe taste. In addition, the acquisition and retrieval of learned associations between novel taste and aversive visceral information requires the activity of aIC-to-BLA projecting neurons (*Kayyal et al., 2019*; *Lavi et al., 2018*). Together, it places the insula as an integrator of two facets of taste saliency; the novel/familiar axis and aversive/appetitive axis. In addition to the necessity of aIC-mPFC interaction in taste novelty behavior and learning, a recent study reported a prominent role of mPFC-to-AIC in learning of novel olfactory tasks (*Zhu et al., 2020*). However, it is yet to be defined if and how the IC-mPFC circuitry encodes saliency for other modalities.

There is a growing body of evidence showing that impairment in entire IC-mPFC circuitry is found in schizophrenia, addiction, and depression (*Cisler et al., 2013*; *Penner et al., 2016*; *Samara et al., 2018*), raising the importance for further investigation of this circuit and its dysfunction in patients. Defining the molecular and cellular mechanisms by which salience of sensory or innate information is processed and defined remains a key question in basic and clinical neuroscience. Our results suggest that while the mPFC plays a critical role in reacting to a novel cue, the aIC is essential both in reacting and in learning of the novel stimulus. This suggests that proposed non-invasive treatments for schizophrenia or addiction (such as transcranial magnetic stimulation therapy) could benefit from regiments that target the IC and/or other salience network components, while the integrity of these networks could be of particular pathognomonic value in the developing and adult brain.

## Acknowledgements

The authors would like to thank all current members of the Rosenblum lab for their help and support, to the veterinary team headed by Barak Carmi and Corina Dollingher and technical team headed by Yair Bellehsen. This research was supported by a grant from the Israel Science Foundation (ISF); ISF 946/17 and ISF 258/20 to KR. HK is a recipient of the Edmond de Rothschild's scholarship. Graphical illustrations were created with BioRender.com.

## Additional information

### Funding

| Funder | Grant reference number | Author |
| --- | --- | --- |
| Israel Science Foundation | isf 946/17 | Kobi Rosenblum |
| Israel Science Foundation | isf 258/20 | Kobi Rosenblum |

The funders had no role in study design, data collection and interpretation, or the decision to submit the work for publication.

### Author contributions

Haneen Kayyal, Conceptualization, Data curation, Formal analysis, Investigation, Methodology, Writing - original draft, Writing - review and editing; Sailendrakumar Kolatt Chandran, Data curation, Formal analysis, Writing - review and editing; Adonis Yiannakas, Mohammad Khamaisy, Data curation, Formal analysis, Investigation, Writing - review and editing; Nathaniel Gould, Data curation, Formal analysis, Investigation, Methodology, Writing - review and editing; Kobi Rosenblum, Conceptualization, Supervision, Funding acquisition, Investigation, Writing - original draft, Writing - review and editing

## Author ORCIDs

Haneen Kayyal ![ORCID] https://orcid.org/0000-0003-4429-3514
Sailendrakumar Kolatt Chandran ![ORCID] http://orcid.org/0000-0002-9805-8096
Kobi Rosenblum ![ORCID] https://orcid.org/0000-0003-4827-0336

## Ethics

Animal experimentation: All experiments and procedures conducted were approved by the University of Haifa Animal Care and Use committee under Ethical license 554/18 and were in accordance with the National Institutes of Health guidelines for ethical treatment of animals.

## Decision letter and Author response

Decision letter https://doi.org/10.7554/eLife.66686.sa1
Author response https://doi.org/10.7554/eLife.66686.sa2

## Additional files

### Supplementary files

• Transparent reporting form

### Data availability

All data generated or analysed during this study are included in the manuscript and supporting files. Source data files have been provided for all figures.

The following datasets were generated:

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
