## [Decision Letter]

**Acceptance summary:**

This work is of interest to neuroscientists in the field of learning and memory as well as those interested in sensory processing and cortico-cortical interactions. It defines reciprocal cortical pathways that differentially support taste novelty behaviour and learning. Overall, the conclusions are well-supported by the data.

**Decision letter after peer review:**

Thank you for submitting your article "Insula to mPFC Reciprocal Connectivity Differentially Underlies Novel Taste Neophobic Response and Learning" for consideration by *eLife*. Your article has been reviewed by 3 peer reviewers, and the evaluation has been overseen by a Reviewing Editor and Laura Colgin as the Senior Editor. The following individuals involved in review of your submission have agreed to reveal their identity: Shauna L Parkes (Reviewer #1); Jelena Radulovic (Reviewer #2); Steven A Kushner (Reviewer #3).

The reviewers have discussed their reviews with one another, and the Reviewing Editor has drafted this to help you prepare a revised submission. Overall, the reviewers found the study interesting and the data provided compelling. The reviewers also pointed the following list of items that you need to address in your revised manuscript. For the revised version, please indicate in the title that the biological model was the mouse in this study (this is an *eLife*'s policy).

Essential revisions:

1. Interpretation of Data

Based on the data provided, is the cortical circuit of interest specific to taste? Can we delineate whether the effective circuit manipulations affect the detection of the novel taste or anxiety associated with presentation of this taste, or both? Providing additional behavioral data (not necessary new experiments but new measures – for example on latency to approach the new taste, number of approaches or total time spent sniffing the bottle – which would indicate novelty detection (or lack thereof) independently of saccharose consumption) or a more thorough discussion of these issues is needed.

2. Include BLA data in the main text

The comparison between IC-to-mPFC versus IC-to-BLA is actually a strength of this manuscript. It supports the specificity of the IC-to-mPFC results and allows us to appreciate the broader circuity at work. Please incorporate the data in the main text (so figures as main figures, not supplements) and discuss them accordingly in the manuscript.

3. Scopolamine data

The scopolamine data were found interesting but with specific issues. The authors could decide to remove them entirely from the manuscript to simplify a take-home at the circuit-level. The authors could also keep the data but they will then need to address some issues:

– Can the authors clarify if increased excitability in mPFC-projecting neurons in the IC truly reflects a learning process?

– Figure 3., an alternative interpretation is that mAchR activation may be required at baseline for maintaining intrinsic properties of aIC neurons (F/I curves for novel vs. familiar compared to saline vs. scopolamine are remarkably similar, the underlying physiological mechanism appears divergent, given the relative discordances in RMP, rheobase, input resistance, and AP half-width [Figures 2/3 and S2/S3]).

– Can we really connect slice data with behavioral, circuit-level data here?

4. Controls

Better CNO controls (especially given differences in size groups) are warranted (controls receiving CNO or in vitro slices).

AAV controls; the selectivity of promoters (cell specificity of infected cells) and cre-dependent constructs (i.e. potential leakage of transgene expression without Cre) need to be examined.

Providing baseline levels for pERK (some areas have elevated baseline activity or could be sensitive to the deprivation protocol) and ideally total numbers of AAV+ cells /layer (in addition to the pErk/AAV %) as this will help estimate the variability of labeling between different animals and the overall number of mPFC-projecting neurons would be much appreciated.

5. Figures

Many figures need to have clearer indication of the layers and areas used for quantification (especially 1h, 4h, sup5b)

*Reviewer #1:*

Kayyal et al. investigated the role of reciprocal pathways between medial prefrontal cortex and insular cortex in novel taste neophobia and memory formation in mice. By combining both correlational (functional neuroanatomy and electrophysiology) and causal techniques (chemogenetics), the authors show that mPFC-projecting neurons in the IC are required for both novel taste expression (neophobia) and taste learning whereas IC-projecting neurons in the mPFC are only involved in the former process. The authors show that consumption of a novel taste increases (1) the number of pERK^+^ neurons in mPFC-projecting neurons in IC and IC-projecting neurons in mPFC, and (2) the excitability of mPFC-projecting neurons in IC, which is also shown to be ACh-dependent. Finally, using a dual-virus chemogenetic approach to target projection-defined neurons, the authors show that inhibition of the IC-to-mPFC pathway hinders both the expression of taste neophobia and its attenuation (i.e., learning). By contrast, the mPFC-to-IC pathway attenuates only the expression of neophobia but not its attenuation. Demonstrating a differential involvement of reciprocally connected cortical regions is innovative and is likely to be of interest to a fairly broad audience in the field of learning and memory.

One of the major strengths of this manuscript is its use of several complementary approaches that, together, inspire a high level of confidence in the results. The authors also include several behavioural and anatomical controls. For instance, the specificity of the IC-to-mPFC in appetitive taste learning is further supported by comparing the involvement of this pathway to the IC-to-BLA pathway, which is shown to be required for taste neophobia but not attenuation of neophobia (i.e., appetitive taste learning).

Overall, the conclusions are well-supported by the data but there are some weaknesses. Most importantly, the extent to which these results are generalizable to other salient stimuli remains unclear and the visual representation and control of the viral expression (for cre-dependent DREADD virus and retro-AAV) is insufficient.

Weaknesses

1. The authors have not sufficiently addressed whether the involvement of this cortical circuitry in salience and novelty processing is strictly limited to taste. The extent to which the reciprocal interactions between mPFC and IC are also needed to "detect, react to, and learn about" other (non-taste related) sensory stimuli remains unclear. While it is understandable that the authors focus their discussion on interactions between IC and mPFC, there is also no discussion (or mention) of the possible involvement of an mPFC-BLA pathway in novel taste behaviour, particularly in the initial expression of a neophobic response. This seems important given the results with quinine (an innately aversive yet still novel taste) indicate that the involvement of the mPFC-IC reciprocal pathways in novel taste behaviour is not ubiquitous.

2. There is no demonstration of the cre-dependency of the AAV construct expressing DREADD at this titre. Also, the visual representation/quantification of the virus expression (both injection sites and retrograde labelling) is insufficient. For example, the injection site of the retro-AAV in Figure 1i appears to predominately target PL but the full extent of the injection site is unclear (same comment for Figure 5a, where M2 also appears to have been targeted). In Figure 4, magnified images of mPFC are shown but we are not provided with a ROI on the map image (Figure 4i). It is also impossible to assess the number of IC projecting neurons in the subregions (PL, Cg, IL). The image provided (Figure 4i) does not actually show IL at this anteroposterior level and there appears to be very little labelling in Cg (although it is difficult to see at this magnification).

3. While an explicit CNO control group is lacking in this paper, any concerns regarding the effect of CNO per se on taste behaviour are somewhat alleviated by the demonstration that CNO injection (inhibition of aIC-mPFC pathway) does not impair taste recognition and retrieval of a familiar taste. Indeed, mice injected with CNO (n=11) but not saline (n=10) show an impaired neophobic response on day 1 (unfamiliar taste) and an increase in the neophobic response on the next (CNO-free) day. When tested on day 7 (when the taste is now familiar), CNO has no effect indicating that recognition and retrieval of a familiar taste is intact. However, on day 7, the group sizes have been reduced to CNO (n=6) and control (n=5). Why are these group sizes different if the same animals were tested? And if these are different mice, why?

4. ACh release in IC is increased following consumption of an unfamiliar taste. The authors showed that novel taste consumption increased excitability in mPFC-projecting neurons in the IC and this excitability was reduced in scopolamine injected mice. However, the authors did not test the mPFC-to-IC pathway. I imagine that the authors would not expect increased excitability in this pathway if its role is limited to expression of neophobia and not learning. But, unfortunately, without this complementary experiment, it remains unclear. Following the same logic, it is unknown if this reduction in excitability is specific to blocking ACh receptors, even if one might expect that it would be ACh-dependent (and NMDA-independent for example) if the increased excitability truly reflects a learning process.

*Reviewer #2:*

While I am in general enthusiastic about this paper, I have several suggestions that I believe might strengthen the manuscript and provide some clarifications regarding the research topic:

1. The key question is related to the overall interpretation that focuses on salience/novelty detection. I found it difficult to delineate whether the effective circuit manipulations affect the detection of the novel taste or anxiety associated with presentation of this taste, or both. I know that these may not be easy to detangle, but I think that consideration of all of these interpretations is needed to fully understand the behavioral effects during expression of neophobia. This can be addressed to some extent by providing additional behavioral data, for example on latency to approach the new taste, number of approaches or total time spent sniffing the bottle, which would indicate novelty detection (or lack thereof) independently of saccharose consumption. Or with a control group receiving a familiar taste (water) in a novel bottle, etc. The authors may already have performed such control experiments, and if so, I would suggest presenting such data to facilitate interpretation. This issue is especially relevant in view of the findings that impaired "expression of neophobia" can nevertheless result in taste memory formation (mPFC to AIC projection results).

2. I would also find it useful to show the baseline levels of pErk (naïve mice) and after water deprivation only in the AIC and mPFC because some areas have elevated baseline activity or could be sensitive to the deprivation protocol. It would also be helpful to see the total numbers of AAV+ cells /layer (in addition to the pErk/AAV %). This will help estimate the variability of labeling between different animals and the overall number of mPFC-projecting neurons.

3. The first figures need to have clear indication of the layers and areas used for quantification.

4. It would be helpful to demonstrate (and validate) the CNO effects in slices.

5. I found two aspects of the work underdeveloped: scopolamine and BLA. I would suggest removing the BLA data from this report because all of it is in the supplement and then suddenly shows in the main Figure denoting the circuit. Strengthening the AIC-mPFC circuit model would suffice in my view. The cholinergic aspect would need much more work to integrate because the slice data are very interesting, yet there are no supporting behavioral or circuit data deepening the understanding of the AIC inner layer cholinergic system in the observed effects. If such link is not easy to establish, I would also tend to omit these data and maybe replace them with the retrieval findings. Alternatively, the discussion on cholinergic mechanisms needs to be much stronger and in tighter connection with the demonstrated circuit.

*Reviewer #3:*

The Rosenblum lab has conducted an important series of experiments to dissect the causal influence of the aIC-to-mPFC connection in distinct aspects of novel taste learning and its adaptive attenuation with subsequent repeated exposure (familiar taste learning). They have used a combination of retrograde labeling to enable visualization of bidirectional monosynaptic connectivity between IC and mPFC, together with pERK labeling to infer in vivo neuronal activation, ex vivo whole cell electrophysiological recordings, and DREADD manipulations to establish the causality of defined connections at distinct cognitive phases during novel taste learning and the subsequent transition to neophobic attenuation with repeated exposure. Overall, the results make an important contribution to our understanding of the cognitive function and plasticity this important neural circuit.

The conclusions of this paper are mostly well supported by data, but a few aspects could be further clarified:

1. AAV8-EF1a and rAAV-hSyn1 promotors for reciprocal aIC-to-mPFC connectivity studies. The EF1a and hSyn1 promoters drive overlapping expression in a broad diversity of neuronal cell types, which leaves open the possibility that distinct classes projection neuron types may be involved. This is particularly relevant for interpretation of the result of the pERK labeling and DREADD inhibition experiments.

2. An alternative explanation for the results in Figure 2 of the novel vs. familiar electrophysiological comparison is that familiar saccharin exposure might decrease intrinsic excitability. Similarly, an alternative explanation for the scopolamine results in Figure 3 is that mAchR activation is required at baseline for maintaining intrinsic properties of aIC neurons. This is especially relevant because although the F/I curves for novel vs. familiar compared to saline vs. scopolamine are remarkably similar, the underlying physiological mechanism appears divergent, given the relative discordances in RMP, rheobase, input resistance, and AP half-width (Figures 2/3 and S2/S3).

3. Previous reports have documented the potential confound on behavioral studies with the use of CNO. Therefore, it would be helpful to confirm whether potential actions of CNO on endogenous neurobiology – independent of its function as a ligand for the DREADD receptor – may be influencing results of key experiments in the manuscript. In particular, it is notable that administration of CNO is associated with a consistent decrease in expression of neophobia across multiple experimental conditions (Figures 5, 6 and S5).

---

## [Author Response]

Essential revisions:1. Interpretation of DataBased on the data provided, is the cortical circuit of interest specific to taste? Can we delineate whether the effective circuit manipulations affect the detection of the novel taste or anxiety associated with presentation of this taste, or both? Providing additional behavioral data (not necessary new experiments but new measures – for example on latency to approach the new taste, number of approaches or total time spent sniffing the bottle – which would indicate novelty detection (or lack thereof) independently of saccharose consumption) or a more thorough discussion of these issues is needed.

We thank the reviewers for this constructive comment.

In our study we tested whether the aIC-mPFC inhibition affects novel taste detection by inhibition of the circuit prior to quinine (novel, innately aversive tastant), which did not affect the neophobic reaction, indicating no changes in novel taste detection.

Following the important comment, we tested whether inhibiting aIC-mPFC pathway affects anxiety levels during an open-field test through testing of exploratory behavior (Seibenhener & Wooten, 2015). Inhibiting aIC-mPFC pathway did not affect the time spent in the center of the arena nor the frequency in crossing the center (new Figure 6—figure supplement 2a, b), indicating no effect on anxiety-related behavior.

Furthermore, we tested the latency towards approaching the novel tastant as suggested by the reviewer, following aIC-mPFC inhibition. No difference was found in latency towards the novel taste nor the water pipettes (new Figure 6—figure supplement 2d,e), indicating no effect on novelty detection or anxiety.

Recent work reported an essential role of mPFC-to-aIC interplay in novel olfactory tasks (Zhu et al., 2020). Although we tested the necessity of aIC-mPFC circuitry in novel taste experience, we do not exclude its involvement in encoding of other sensory stimuli as suggested by the reviewer. Hence, we addressed this important issue in the Discussion section.

2. Include BLA data in the main textThe comparison between IC-to-mPFC versus IC-to-BLA is actually a strength of this manuscript. It supports the specificity of the IC-to-mPFC results and allows us to appreciate the broader circuity at work. Please incorporate the data in the main text (so figures as main figures, not supplements) and discuss them accordingly in the manuscript.

We appreciate the reviewer’s comment, which has led us to greatly improve the flow and clarity of the paper. BLA data was incorporated to the main text as suggested by the reviewers.

3. Scopolamine dataThe scopolamine data were found interesting but with specific issues. The authors could decide to remove them entirely from the manuscript to simplify a take-home at the circuit-level. The authors could also keep the data but they will then need to address some issues:– Can the authors clarify if increased excitability in mPFC-projecting neurons in the IC truly reflects a learning process?– Figure 3., an alternative interpretation is that mAchR activation may be required at baseline for maintaining intrinsic properties of aIC neurons (F/I curves for novel vs. familiar compared to saline vs. scopolamine are remarkably similar, the underlying physiological mechanism appears divergent, given the relative discordances in RMP, rheobase, input resistance, and AP half-width [Figures 2/3 and S2/S3]).– Can we really connect slice data with behavioral, circuit-level data here?

We thank the reviewer for raising this important issue, and accordingly following debating, we removed the scopolamine data from the manuscript.

Slice data has its known limitations as mentioned by the reviewer. However, previous published slice electrophysiology data (Bloodgood et al., 2018; Campanac et al., 2013; Dunn et al., 2019; Otis et al., 2018; Pignatelli et al., 2019; Sano et al., 2014; Santini et al., 2008; Santini & Porter, 2010; Sehgal et al., 2014; Soler-Cedeño et al., 2016; Song et al., 2015; Viosca et al., 2009; Wayman & Woodward, 2018; Whitaker et al., 2017; Yiannakas et al., 2021; Yiu et al., 2014), showed an important correlation between circuit level changes and increased excitability and intrinsic properties in specific neurons, following learning processes. Importantly, we are aware of the limitations and continued the examination using causative relation in the behaving mice.

4. ControlsBetter CNO controls (especially given differences in size groups) are warranted (controls receiving CNO or in vitro slices).AAV controls; the selectivity of promoters (cell specificity of infected cells) and cre-dependent constructs (i.e. potential leakage of transgene expression without Cre) need to be examined.Providing baseline levels for pERK (some areas have elevated baseline activity or could be sensitive to the deprivation protocol) and ideally total numbers of AAV+ cells /layer (in addition to the pErk/AAV %) as this will help estimate the variability of labeling between different animals and the overall number of mPFC-projecting neurons would be much appreciated.

We thank the reviewers for highlighting the need of further extension of our control experiments. In response to these important points raised by the reviewers, we performed additional experiments.

In order to address how CNO administration by itself affect classical neophobia or its attenuation, control mice (injected with retro-Cre AAV construct at the mPFC, and AAV8_hEF1a-dlox-mCherry (rev)-dlox-WPRE-hGHp (A), an mCherry construct without DREADD receptor, at the IC.) were prepared, and similar behavioral taste-tests were performed. There was no difference in aversion towards novel tastant (or its attenuation) following CNO administration in control-virus injected mice and the saline injected group (new Figure 4—figure supplement 1a). This result indicates no effect of CNO in taste behavior (in the concentration we used).

Regarding cell-specificity of infected cells: Indeed, different neuronal populations (excitatory and inhibitory) might have been targeted by our viral vector containing the synapsin promoter. However, from the literature (Baker et al., 2018; Gehrlach et al., 2020; Greig et al., 2013; Molyneaux et al., 2007; Parnavelas, 2000), we know that most cortical projections (>95%) are excitatory neurons. In addition, and in parallel, this is also notable in our electrophysiological measurements by which all neurons patched show a pyramidal-neuron-like properties. Hence, the probability of targeting inhibitory neurons is low to nothing.

As for the concern regarding potential leakage of transgene expression without Cre: we have used all cre-dependent virus -double inverted floxed (AAV8_hEF1a-dlox-hM4D(Gi)_mCherry(rev)-dlox-WPRE-hGHp(A), aiming for high specificity to cre recombination and less likely to the leakage. The expression of the target gene is specific in cells that express Cre recombinase protein. (Tervo et al., 2016; Zerbi et al., 2019).

Following the reviewers' comment and in order to further clarify the aIC-mPFC anatomical connectivity, we measured and provided data for total numbers of rAAV^+^ cells in the different subregions/layers (Figure 1—figure supplementary 1a-d). Number of rAAV+ neurons in the aIC and the mPFC was similar both in novel and familiar saccharin groups, indicating no variability in injections/labeling between the groups.

5. FiguresMany figures need to have clearer indication of the layers and areas used for quantification (especially 1h, 4h, sup5b)

As suggested by the reviewer, we amended the figures to improve clarity. Clearer indication of the areas and layers examined was provided in the figures.

Reviewer #1:Kayyal et al. investigated the role of reciprocal pathways between medial prefrontal cortex and insular cortex in novel taste neophobia and memory formation in mice. By combining both correlational (functional neuroanatomy and electrophysiology) and causal techniques (chemogenetics), the authors show that mPFC-projecting neurons in the IC are required for both novel taste expression (neophobia) and taste learning whereas IC-projecting neurons in the mPFC are only involved in the former process. The authors show that consumption of a novel taste increases (1) the number of pERK^+^ neurons in mPFC-projecting neurons in IC and IC-projecting neurons in mPFC, and (2) the excitability of mPFC-projecting neurons in IC, which is also shown to be ACh-dependent. Finally, using a dual-virus chemogenetic approach to target projection-defined neurons, the authors show that inhibition of the IC-to-mPFC pathway hinders both the expression of taste neophobia and its attenuation (i.e., learning). By contrast, the mPFC-to-IC pathway attenuates only the expression of neophobia but not its attenuation. Demonstrating a differential involvement of reciprocally connected cortical regions is innovative and is likely to be of interest to a fairly broad audience in the field of learning and memory.One of the major strengths of this manuscript is its use of several complementary approaches that, together, inspire a high level of confidence in the results. The authors also include several behavioural and anatomical controls. For instance, the specificity of the IC-to-mPFC in appetitive taste learning is further supported by comparing the involvement of this pathway to the IC-to-BLA pathway, which is shown to be required for taste neophobia but not attenuation of neophobia (i.e., appetitive taste learning).Overall, the conclusions are well-supported by the data but there are some weaknesses. Most importantly, the extent to which these results are generalizable to other salient stimuli remains unclear and the visual representation and control of the viral expression (for cre-dependent DREADD virus and retro-AAV) is insufficient.Weaknesses1. The authors have not sufficiently addressed whether the involvement of this cortical circuitry in salience and novelty processing is strictly limited to taste. The extent to which the reciprocal interactions between mPFC and IC are also needed to "detect, react to, and learn about" other (non-taste related) sensory stimuli remains unclear. While it is understandable that the authors focus their discussion on interactions between IC and mPFC, there is also no discussion (or mention) of the possible involvement of an mPFC-BLA pathway in novel taste behaviour, particularly in the initial expression of a neophobic response. This seems important given the results with quinine (an innately aversive yet still novel taste) indicate that the involvement of the mPFC-IC reciprocal pathways in novel taste behaviour is not ubiquitous.

We sincerely thank the reviewer for their valuable comments and suggestions.

Both the IC and the mPFC are heavily implicated in learning of novel sensory stimuli (Bermudez-Rattoni et al., 2005; Gogolla, 2017; Morici et al., 2015; Wang et al., 2020) and the connectivity between them was found to be crucial in learning novel olfactory tasks (Zhu et al., 2020). This indeed raises the possible involvement of aIC-mPFC circuitry in reacting-to and learning of sensory modalities other than taste learning, which we addressed in the Discussion section as suggested by the reviewer.

The mPFC and the BLA are reciprocally and functionally connected (Yizhar & Klavir, 2018). This interplay has been shown to be crucial for learning of valence-specific behaviors (Yizhar & Klavir, 2018) including water intake (Zhao et al., 2020) and in innate fear responses (Jhang et al., 2018). Indeed, it is reasonable to suggest that the interplay between the BLA and the mPFC plays a role in novelty detection and learning as mentioned by the reviewer, and hence we mentioned it in the discussion.

A more thorough debate of the interplay between the BLA and the mPFC was added to the discussion, as well as the involvement of the aIC-mPFC circuit in other modalities.

2. There is no demonstration of the cre-dependency of the AAV construct expressing DREADD at this titre. Also, the visual representation/quantification of the virus expression (both injection sites and retrograde labelling) is insufficient. For example, the injection site of the retro-AAV in Figure 1i appears to predominately target PL but the full extent of the injection site is unclear (same comment for Figure 5a, where M2 also appears to have been targeted). In Figure 4, magnified images of mPFC are shown but we are not provided with a ROI on the map image (Figure 4i). It is also impossible to assess the number of IC projecting neurons in the subregions (PL, Cg, IL). The image provided (Figure 4i) does not actually show IL at this anteroposterior level and there appears to be very little labelling in Cg (although it is difficult to see at this magnification).

We thank the reviewers for raising this important issue. We agree that electrophysiological demonstration and quantification of the effects of CNO in slices expressing inhibitory DREADDs is fundamental to the credibility of the results outlined in the manuscript. We have previously shown a proof of concept for our viral tools in a study published previously (Kayyal et al., 2019) using the rAAV Cre and the cre-dependent DREADDs in IC-to-BLA projecting neurons.

As recommended by the reviewer, we have included overlays confirming that our injection was confined within the aIC or mPFC (Figure 1 and Figure 3), in inhibitory DREADD as well as in correlative pERK experiments. We believe that the above indeed improves the clarity of the presented images as suggested by the reviewer.

3. While an explicit CNO control group is lacking in this paper, any concerns regarding the effect of CNO per se on taste behaviour are somewhat alleviated by the demonstration that CNO injection (inhibition of aIC-mPFC pathway) does not impair taste recognition and retrieval of a familiar taste. Indeed, mice injected with CNO (n=11) but not saline (n=10) show an impaired neophobic response on day 1 (unfamiliar taste) and an increase in the neophobic response on the next (CNO-free) day. When tested on day 7 (when the taste is now familiar), CNO has no effect indicating that recognition and retrieval of a familiar taste is intact. However, on day 7, the group sizes have been reduced to CNO (n=6) and control (n=5). Why are these group sizes different if the same animals were tested? And if these are different mice, why?

Though we used a concentration of CNO considered to be specific, we thank the reviewers for this constructive comment and preform an additional experiment to validate it. Hence, we added a control group, in which mice were injected with retro-Cre AAV construct at the mPFC, and AAV8_hEF1a-dlox-mCherry (rev)-dlox-WPRE-hGHp (A), an mCherry construct without DREADD receptor, at the aIC. Animals were injected with CNO prior to novel taste procedure (Figure 4—figure supplementary 1a) to control for CNO effects. We observed no difference between the control-virus injected mice that were treated with CNO and the aIC-to-mPFC inhibitory DREADDs-injected mice. This rules out the possible involvement of CNO itself on the measured taste behavior.

Regarding the changes in number of mice in the different procedures- The first experiment was done with an n number of (CNO=5, saline=5), without testing the effect on retrieval of familiar taste. In the second batch, we used another set of animals (CNO=6, saline=5), where we wanted to increase the n number of our experiment, keeping in mind we wanted to control this time for taste *recognition*. Since, there is no difference between the groups, we did not repeat this test.

4. ACh release in IC is increased following consumption of an unfamiliar taste. The authors showed that novel taste consumption increased excitability in mPFC-projecting neurons in the IC and this excitability was reduced in scopolamine injected mice. However, the authors did not test the mPFC-to-IC pathway. I imagine that the authors would not expect increased excitability in this pathway if its role is limited to expression of neophobia and not learning. But, unfortunately, without this complementary experiment, it remains unclear. Following the same logic, it is unknown if this reduction in excitability is specific to blocking ACh receptors, even if one might expect that it would be ACh-dependent (and NMDA-independent for example) if the increased excitability truly reflects a learning process.

We thank the reviewer for pointing this out. Accordingly, we have removed all the scopolamine data from the manuscript. The manuscript is now better focused on the main hypothesis and take-home messages are clearer.

It would have been interesting to explore the changes occurring in the mPFC following novel taste experience, however, it is not within the scope of our study. The main focus in our paper was to better understand the changes in the aIC and the circuits involving it, in novel taste experience.

Reviewer #2:While I am in general enthusiastic about this paper, I have several suggestions that I believe might strengthen the manuscript and provide some clarifications regarding the research topic:1. The key question is related to the overall interpretation that focuses on salience/novelty detection. I found it difficult to delineate whether the effective circuit manipulations affect the detection of the novel taste or anxiety associated with presentation of this taste, or both. I know that these may not be easy to detangle, but I think that consideration of all of these interpretations is needed to fully understand the behavioral effects during expression of neophobia. This can be addressed to some extent by providing additional behavioral data, for example on latency to approach the new taste, number of approaches or total time spent sniffing the bottle, which would indicate novelty detection (or lack thereof) independently of saccharose consumption. Or with a control group receiving a familiar taste (water) in a novel bottle, etc. The authors may already have performed such control experiments, and if so, I would suggest presenting such data to facilitate interpretation. This issue is especially relevant in view of the findings that impaired "expression of neophobia" can nevertheless result in taste memory formation (mPFC to AIC projection results).

We would like to thank the reviewer for their important feedback and for bringing this issue to our attention.

To address the reviewer’s comment, we sought to test the effect of the aIC-mPFC inhibition on anxiety levels. We targeted aIC-mPFC pathway, and tested the open-field behavior when the pathway is manipulated similarly to other experiments in the manuscript. The amount of time spent in the center or corners of the chamber provide us a measurement of stress and anxiety levels during inhibition of aIC-mPFC circuitry (new Figure 6—figure supplementary 2a-c). Furthermore, we tested the latency towards the novel tastant during aIC-mPFC inhibition as suggested by the reviewer. Our results show that inhibiting aIC-mPFC circuit does not affect anxiety levels as measured in the open field test nor in latency towards the novel taste tested (new Figure 6—figure supplementary 2a-e).

2. I would also find it useful to show the baseline levels of pErk (naïve mice) and after water deprivation only in the AIC and mPFC because some areas have elevated baseline activity or could be sensitive to the deprivation protocol. It would also be helpful to see the total numbers of AAV+ cells /layer (in addition to the pErk/AAV %). This will help estimate the variability of labeling between different animals and the overall number of mPFC-projecting neurons.

We thank the reviewers for the suggestions aimed to extend our analysis.

Indeed, drinking following water restriction affects gene expression such as Arc and cFos in the IC (Inberg et al., 2016). However, in order to control the baseline levels of pERK, both novel and familiar taste groups were water restricted equally during the behavioral paradigm. Also, mice were provided with an equal amount (1 ml) of saccharin in order to prevent variability in volumes consumed in the last day. This ensures the equal effect of dehydration/drinking in both groups and allows comparison of the familiarity-novelty axis only.

Following the reviewers' comment and in order to further clarify the aIC-mPFC anatomical connectivity, we performed quantification of rAAV^+^ neurons in the aIC and the mPFC and provided the data as a supplementary figure (Figure 1—figure supplement1) to better clarify the variability of labeling between the groups. Number of rAAV+ neurons in the aIC and the mPFC was similar both in novel and familiar saccharin groups, indicating no variability in injections/labeling between the groups.

3. The first figures need to have clear indication of the layers and areas used for quantification.

Thanking the reviewers for the comment. We amended the figures to improve clarity as proposed.

4. It would be helpful to demonstrate (and validate) the CNO effects in slices.

The reviewer raises an important point. We agree that electrophysiological demonstration and quantification of the effects of CNO in slices expressing inhibitory DREADDs is fundamental to the credibility of the results. We have previously shown and validated the effect of CNO in slices (Kayyal et al., 2019) in aIC-to-BLA projecting neurons, using *the same* viral vectors (rAAV-Cre at the BLA and Cre-dependent DREADDs in aIC).

5. I found two aspects of the work underdeveloped: scopolamine and BLA. I would suggest removing the BLA data from this report because all of it is in the supplement and then suddenly shows in the main Figure denoting the circuit. Strengthening the AIC-mPFC circuit model would suffice in my view. The cholinergic aspect would need much more work to integrate because the slice data are very interesting, yet there are no supporting behavioral or circuit data deepening the understanding of the AIC inner layer cholinergic system in the observed effects. If such link is not easy to establish, I would also tend to omit these data and maybe replace them with the retrieval findings. Alternatively, the discussion on cholinergic mechanisms needs to be much stronger and in tighter connection with the demonstrated circuit.

We thank the reviewers for bringing this idea to our notice. To improve the flow and clarity of the paper, BLA data was incorporated to the main text as suggested by the reviewers. Additionally, scopolamine data was removed from the manuscript.

Reviewer #3:The Rosenblum lab has conducted an important series of experiments to dissect the causal influence of the aIC-to-mPFC connection in distinct aspects of novel taste learning and its adaptive attenuation with subsequent repeated exposure (familiar taste learning). They have used a combination of retrograde labeling to enable visualization of bidirectional monosynaptic connectivity between IC and mPFC, together with pERK labeling to infer in vivo neuronal activation, ex vivo whole cell electrophysiological recordings, and DREADD manipulations to establish the causality of defined connections at distinct cognitive phases during novel taste learning and the subsequent transition to neophobic attenuation with repeated exposure. Overall, the results make an important contribution to our understanding of the cognitive function and plasticity this important neural circuit.The conclusions of this paper are mostly well supported by data, but a few aspects could be further clarified:1. AAV8-EF1a and rAAV-hSyn1 promotors for reciprocal aIC-to-mPFC connectivity studies. The EF1a and hSyn1 promoters drive overlapping expression in a broad diversity of neuronal cell types, which leaves open the possibility that distinct classes projection neuron types may be involved. This is particularly relevant for interpretation of the result of the pERK labeling and DREADD inhibition experiments.

We thank the reviewer for pointing this out. We agree with this comment, while hSyn1 promoter is directed only to neuronal population, EF1a directs a broader cell population. However, in the causative experiments in which we inhibited either aIC-to-mPFC or mPFC-to-aIC circuit, we used a retro-Cre AAV construct: rAAV-hSyn1-chI-EGFP_2A_iCre-WPRE-SV40p(A). meaning the targeting eventually was similar (hSyn1 promoter) both in the correlative and the causative studies.

We know from the literature (Baker et al., 2018; Gehrlach et al., 2020; Greig et al., 2013; Molyneaux et al., 2007; Parnavelas, 2000) that cortical and subcortical projecting neurons are mostly excitatory and not inhibitory neurons (>95%). In addition, our electrophysiological studies show that all neurons patched have pyramidal-neuron-like properties. Hence, the probability of us targeting inhibitory neurons is very small.

2. An alternative explanation for the results in Figure 2 of the novel vs. familiar electrophysiological comparison is that familiar saccharin exposure might decrease intrinsic excitability. Similarly, an alternative explanation for the scopolamine results in Figure 3 is that mAchR activation is required at baseline for maintaining intrinsic properties of aIC neurons. This is especially relevant because although the F/I curves for novel vs. familiar compared to saline vs. scopolamine are remarkably similar, the underlying physiological mechanism appears divergent, given the relative discordances in RMP, rheobase, input resistance, and AP half-width (Figures 2/3 and S2/S3).

In response to this important point raised by the reviewer, we have performed an additional experiment to examine the directionality of novel/familiar taste experience upon aIC-to-mPFC projecting neurons by adding a home-caged group of mice (Figure 2; Figure 2—figure supplement 2). Our results show a similar excitability levels between home-caged group and familiar saccharin group that are different from novel saccharin group, meaning the effect is derived from the novelty of the taste (increase excitability in aIC-to-mPFC projections).

In addition, the scopolamine data was removed as suggested by the reviewers.

3. Previous reports have documented the potential confound on behavioral studies with the use of CNO. Therefore, it would be helpful to confirm whether potential actions of CNO on endogenous neurobiology – independent of its function as a ligand for the DREADD receptor – may be influencing results of key experiments in the manuscript. In particular, it is notable that administration of CNO is associated with a consistent decrease in expression of neophobia across multiple experimental conditions (Figures 5, 6 and S5).

We thank the reviewer for raising this important concern.

Doses of CNO used in all experiments were chosen based on recent publications demonstrating that higher chronic doses of the ligand are metabolized into clozapine, which can affect behavior (Gomez et al., 2017).

To directly address whether CNO administration by itself affect classical neophobia or its attenuation, we performed an additional experiment. Control mice (injected with control virus without DREADDs expression at the IC) were prepared, and similar behavioral taste-tests were performed. Our results indicate that the expression of neophobia is unchanged by CNO administration when using control virus (which does not harbor the receptor). This rule out the possibility that the measured behavior was affected by the CNO itself (Figure 4—figure supplement 1a).

References:

Baker, A., Kalmbach, B., Morishima, M., Kim, J., Juavinett, A., Li, N., and Dembrow, N. (2018). Specialized subpopulations of deep-layer pyramidal neurons in the neocortex: Bridging cellular properties to functional consequences. Journal of Neuroscience, 38(24), 5441–5455. https://doi.org/10.1523/JNEUROSCI.0150-18.2018

Bermudez-Rattoni, F., Okuda, S., Roozendaal, B., and McGaugh, J. L. (2005). Insular cortex is involved in consolidation of object recognition memory. Learning and Memory, 12(5), 447–449. https://doi.org/10.1101/lm.97605

Bloodgood, D. W., Sugam, J. A., Holmes, A., and Kash, T. L. (2018). Fear extinction requires infralimbic cortex projections to the basolateral amygdala. Translational Psychiatry, 8(1). https://doi.org/10.1038/s41398-018-0106-x

Campanac, E., Gasselin, C., Baude, A., Rama, S., Ankri, N., and Debanne, D. (2013). Enhanced Intrinsic Excitability in Basket Cells Maintains Excitatory-Inhibitory Balance in Hippocampal Circuits. Neuron, 77(4), 712–722. https://doi.org/10.1016/j.neuron.2012.12.020

Dunn, A. R., Neuner, S. M., Ding, S., Hope, K. A., M.S., K. O., and Kaczorowski, C. C. (2019). Cell-Type-Specific Changes in Intrinsic Excitability in the Subiculum following Learning and Exposure to Novel Environmental Contexts. ENeuro, 5(6), 18.2018. https://doi.org/10.1523/ENEURO.0484-18.2018

Gehrlach, D. A., Weiand, C., Gaitanos, T. N., Cho, E., Klein, A. S., Hennrich, A. A., Conzelmann, K. K., and Gogolla, N. (2020). A whole-brain connectivity map of mouse insular cortex. *eLife*, 9, 1–78. https://doi.org/10.7554/*eLife*.55585

Gogolla, N. (2017). The insular cortex. In Current Biology (Vol. 27, Issue 12, pp. R580–R586). Cell Press. https://doi.org/10.1016/j.cub.2017.05.010

Gomez, J. L., Bonaventura, J., Lesniak, W., Mathews, W. B., Sysa-Shah, P., Rodriguez, L. A., Ellis, R. J., Richie, C. T., Harvey, B. K., Dannals, R. F., Pomper, M. G., Bonci, A., and Michaelides, M. (2017). Chemogenetics revealed: DREADD occupancy and activation via converted clozapine. Science, 357(6350), 503–507. https://doi.org/10.1126/science.aan2475

Greig, L. C., Woodworth, M. B., Galazo, M. J., Padmanabhan, H., and Macklis, J. D. (2013). Molecular logic of neocortical projection neuron specification, development and diversity. NATURE REVIEWS | NEUROSCIENCE, 14, 755. https://doi.org/10.1038/nrn3586

Inberg, S., Jacob, E., Elkobi, A., Edry, E., Rappaport, A., Simpson, T. I., Armstrong, J. D., Shomron, N., Pasmanik-Chor, M., and Rosenblum, K. (2016). Fluid consumption and taste novelty determines transcription temporal dynamics in the gustatory cortex. Molecular Brain, 9(1). https://doi.org/10.1186/s13041-016-0188-4

Jhang, J., Lee, H., Kang, M. S., Lee, H. S., Park, H., and Han, J. H. (2018). Anterior cingulate cortex and its input to the basolateral amygdala control innate fear response. Nature Communications, 9(1). https://doi.org/10.1038/s41467-018-05090-y

Kayyal, H., Yiannakas, A., Kolatt Chandran, S., Khamaisy, M., Sharma, V., and Rosenblum, K. (2019). Activity of Insula to Basolateral Amygdala Projecting Neurons is Necessary and Sufficient for Taste Valence Representation. The Journal of Neuroscience : The Official Journal of the Society for Neuroscience, 39(47), 9369–9382. https://doi.org/10.1523/JNEUROSCI.0752-19.2019

Molyneaux, B. J., Arlotta, P., L Menezes, J. R., and Macklis, J. D. (2007). Neuronal subtype specification in the cerebral cortex. NATURE REVIEWS | NEUROSCIENCE, 8. https://doi.org/10.1038/nrn2151

Morici, J. F., Bekinschtein, P., and Weisstaub, N. V. (2015). Medial prefrontal cortex role in recognition memory in rodents. In Behavioural Brain Research (Vol. 292, pp. 241–251). Elsevier. https://doi.org/10.1016/j.bbr.2015.06.030

Otis, J. M., Fitzgerald, M. K., Yousuf, H., Burkard, J. L., Drake, M., and Mueller, D. (2018). Prefrontal neuronal excitability maintains cocaine-associated memory during retrieval. Frontiers in Behavioral Neuroscience, 12(June), 1–11. https://doi.org/10.3389/fnbeh.2018.00119

Parnavelas, J. G. (2000). The origin and migration of cortical neurones: New vistas. In Trends in Neurosciences (Vol. 23, Issue 3, pp. 126–131). Trends Neurosci. https://doi.org/10.1016/S0166-2236(00)01553-8

Pignatelli, M., Ryan, T. J., Roy, D. S., Lovett, C., Smith, L. M., Muralidhar, S., and Tonegawa, S. (2019). Engram Cell Excitability State Determines the Efficacy of Memory Retrieval. Neuron, 101(2), 274-284.e5. https://doi.org/10.1016/j.neuron.2018.11.029

Sano, Y., Shobe, J. L., Zhou, M., Huang, S., Shuman, T., Cai, D. J., Golshani, P., Kamata, M., and Silva, A. J. (2014). CREB regulates memory allocation in the insular cortex. Current Biology, 24(23), 2833–2837. https://doi.org/10.1016/j.cub.2014.10.018

Santini, E., and Porter, J. T. (2010). M-Type Potassium Channels Modulate the Intrinsic Excitability of Infralimbic Neurons and Regulate Fear Expression and Extinction. 30(37), 12379–12386.

Santini, E., Quirk, G. J., and Porter, J. T. (2008). Fear Conditioning and Extinction Differentially Modify the Intrinsic Excitability of Infralimbic Neurons. Journal of Neuroscience, 28(15), 4028–4036.

Sehgal, M., Ehlers, V. L., and Moyer James R, J. (2014). Learning enhances intrinsic excitability in a subset of lateral amygdala neurons. Learning and Memory (Cold Spring Harbor, N.Y.), 21(3), 161–170. https://doi.org/10.1101/lm.032730.113

Seibenhener, M. L., and Wooten, M. C. (2015). Use of the open field maze to measure locomotor and anxiety-like behavior in mice. Journal of Visualized Experiments, 96. https://doi.org/10.3791/52434

Soler-Cedeño, O., Cruz, E., Criado-Marrero, M., and Porter, J. T. (2016). Contextual fear conditioning depresses infralimbic excitability. Neurobiology of Learning and Memory, 130(February), 77–82. https://doi.org/10.1016/j.nlm.2016.01.015

Song, C., Ehlers, V. L., Moyer, J. R., and R, J. M. J. (2015). Trace Fear Conditioning Differentially Modulates Intrinsic Excitability of Medial Prefrontal Cortex-Basolateral Complex of Amygdala Projection Neurons in Infralimbic and Prelimbic Cortices. 35(39), 13511–13524.

Tervo, D. G. R., Hwang, B. Y., Viswanathan, S., Gaj, T., Lavzin, M., Ritola, K. D., Lindo, S., Michael, S., Kuleshova, E., Ojala, D., Huang, C. C., Gerfen, C. R., Schiller, J., Dudman, J. T., Hantman, A. W., Looger, L. L., Schaffer, D. V., and Karpova, A. Y. (2016). A Designer AAV Variant Permits Efficient Retrograde Access to Projection Neurons. Neuron, 92(2), 372–382. https://doi.org/10.1016/j.neuron.2016.09.021

Viosca, J., De Armentia, M. L., Jancic, D., and Barco, A. (2009). Enhanced CREB-dependent gene expression increases the excitability of neurons in the basal amygdala and primes the consolidation of contextual and cued fear memory. Learning and Memory, 16(3), 193–197. https://doi.org/10.1101/lm.1254209

Wang, P. Y., Boboila, C., Chin, M., Higashi-Howard, A., Shamash, P., Wu, Z., Stein, N. P., Abbott, L. F., and Axel, R. (2020). Transient and Persistent Representations of Odor Value in Prefrontal Cortex. Neuron, 108(1), 209-224.e6. https://doi.org/10.1016/j.neuron.2020.07.033

Wayman, W. N., and Woodward, J. J. (2018). Chemogenetic excitation of accumbens-projecting infralimbic cortical neurons blocks toluene-induced conditioned place preference. Journal of Neuroscience, 38(6), 1462–1471. https://doi.org/10.1523/JNEUROSCI.2503-17.2018

Whitaker, L. R., Warren, B. L., Venniro, M., Harte, T. C., McPherson, K. B., Beidel, J., Bossert, J. M., Shaham, Y., Bonci, A., and Hope, B. T. (2017). Bidirectional modulation of intrinsic excitability in rat prelimbic cortex neuronal ensembles and non-ensembles after operant learning. Journal of Neuroscience, 37(36), 8845–8856. https://doi.org/10.1523/JNEUROSCI.3761-16.2017

Yiannakas, A., Kolatt Chandran, S., Kayyal, H., Gould, N., Khamaisy, M., and Rosenblum, K. (2021). Parvalbumin interneuron inhibition onto anterior insula neurons projecting to the basolateral amygdala drives aversive taste memory retrieval. Current Biology, 1–15. https://doi.org/10.1016/j.cub.2021.04.010

Yiu, A. P., Mercaldo, V., Yan, C., Richards, B., Rashid, A. J., Hsiang, H. L. L., Pressey, J., Mahadevan, V., Tran, M. M., Kushner, S. A., Woodin, M. A., Frankland, P. W., and Josselyn, S. A. (2014). Neurons Are Recruited to a Memory Trace Based on Relative Neuronal Excitability Immediately before Training. Neuron, 83(3), 722–735. https://doi.org/10.1016/j.neuron.2014.07.017

Yizhar, O., and Klavir, O. (2018). Reciprocal amygdala–prefrontal interactions in learning. In Current Opinion in Neurobiology (Vol. 52, pp. 149–155). Elsevier Ltd. https://doi.org/10.1016/j.conb.2018.06.006

Zerbi, V., Floriou-Servou, A., Markicevic, M., Vermeiren, Y., Sturman, O., Privitera, M., von Ziegler, L., Ferrari, K. D., Weber, B., De Deyn, P. P., Wenderoth, N., and Bohacek, J. (2019). Rapid Reconfiguration of the Functional Connectome after Chemogenetic Locus Coeruleus Activation. Neuron, 103(4), 702-718.e5. https://doi.org/10.1016/j.neuron.2019.05.034

Zhao, Z., Soria-Gómez, E., Varilh, M., Covelo, A., Julio-Kalajzić, F., Cannich, A., Castiglione, A., Vanhoutte, L., Duveau, A., Zizzari, P., Beyeler, A., Cota, D., Bellocchio, L., Busquets-Garcia, A., and Marsicano, G. (2020). A Novel Cortical Mechanism for Top-Down Control of Water Intake. Current Biology, 30(23), 4789-4798.e4. https://doi.org/10.1016/j.cub.2020.09.011

Zhu, J., Cheng, Q., Chen, Y., Fan, H., Han, Z., Hou, R., Chen, Z., and Li, C. T. (2020). Transient Delay-Period Activity of Agranular Insular Cortex Controls Working Memory Maintenance in Learning Novel Tasks. Neuron, 105(5), 934-946.e5. https://doi.org/10.1016/j.neuron.2019.12.008